# Mechanical behavior of synthetic fiber ropes for mooring floating offshore wind turbines

Ji Zeng[1], He Zhang [2]*, Bowen Jin[2], Hailei Dong[3], Chiate Chou[3], Hangyu Li[3]

1 College of Ocean Science and Engineering, Shanghai Maritime University, Shanghai, China, 2 Marine College, Shanghai Maritime University, Shanghai, China, 3 Zhejiang Four Brothers Rope Co., Ltd., Taizhou, Zhejiang, China

* zhanghe_mu@163.com

**Data Availability Statement:** All relevant data are within the manuscript and its Supporting Information files.

## Abstract

The continuous development of floating wind turbine technology is pushing platforms further offshore into deeper seas. Consequently, the water depth required for mooring is increasing, demanding higher standards for the mooring system. This study explores the mechanical properties, including wear resistance and quasi-static stiffness, of nylon, polyester, and high-strength polyethylene mooring ropes through experimental design, aiming to address the challenges faced by floating offshore wind turbines (FOWT) in mooring line safety. In conducting yarn-on-yarn abrasion tests, materials, twist, marine lubricants, and dry and wet environments were chosen as research variables to analyse and compare the wear and tear frequency of sample ropes. The study found that the degree of wear on ropes composed of different materials is affected differently by twist and dry and wet environments, and the application of marine lubricants can significantly extend the friction fracture cycle of yarns. To study the tensile properties and static stiffness characteristics of polyester and nylon ropes, the cables were divided into three stages: preloading, initial installation, and ageing. The strain and reversible elongation of the cables at each stage were analysed, and an empirical formula for quasi-static stiffness considering the creep coefficient of the ropes was established. The study found that the ropes had reversible elongation after being left stationary, the more thorough the running-in, the smaller the inherent deformation, and the more stable the structure. The static stiffness of the cables increased with the loading time and force; however, it eventually approached a constant value, and the stiffness of polyester cables was found to be greater than that of nylon ropes. These results provide a solid theoretical foundation and practical guidance for the material selection of FOWT cables, which contributes to advancing the application and development of FOWT in renewable energy.

## 1. Introduction

Offshore wind power can harness offshore wind resources to generate clean energy, reduce dependence on traditional fossil fuels and carbon emissions, and address climate change. The increasing demand for renewable energy has led to extensive research on floating offshore wind turbines (FOWTs) as a potential clean energy generation technology. Compared to

**Funding:** The author(s) received no specific funding for this work.

**Competing interests:** The authors have declared that no competing interests exist.

traditional fixed wind turbines, FOWTs yield a higher power generation efficiency and offer a broader range of applications. However, owing to the distinctive nature of the environment, which is constantly affected by factors such as wind and waves, the design of the mooring system and mechanical performance of FOWTs have become important research directions. As a key component of an FOWT mooring system, the cable plays a crucial role in ensuring the stable operation and safety of the equipment. Therefore, selecting appropriate cable materials and understanding the mechanical characteristics of synthetic fibre cables are crucial factors in ensuring the reliability and stability of FOWTs.

As FOWT platforms develop toward offshore locations, the mooring water depth increases, the length of the mooring chain line grows, and the weight of the anchor chain becomes a burden. Metallic chain lines cannot fulfil the increasing development needs of offshore floating platforms. Owing to the differences in mooring materials and structures, they exhibit different advantages and disadvantages in terms of mechanical properties [1]. Synthetic fibre cable is a mooring line made of woven fibre materials, with a density usually lower than 1.5 g/cm$^3$, matching the buoyancy of seawater, thereby effectively reducing the weight of the mooring line [2]. As early as the 1960s, scholars proposed using synthetic fibres in deepwater taut mooring systems [3]. Compared to traditional catenary mooring, fibre ropes can reduce the anchor line radius by 40%, thereby increasing the effective payload and horizontal response efficiency of the float [4]. However, compared to steel ropes, fibre materials comprising molecules with viscoelastic properties exhibit hysteresis, damping, and other features [5]. The tensile behavior of ropes exhibits nonlinear characteristics influenced by factors such as load size and frequency, and it depends on the load action duration [6], which increases the difficulty of analysing and verifying the mechanical properties of ropes.

During usage, ropes experience progressive wear and damage to their surface and internal microstructure owing to cyclic loading forces, leading to fibre breakage and loosening. As the number of cycles increases, wear gradually extends to a wider range and depth. Polyester fibre ropes have been used for deep-sea mooring for many years [7], and extensive fatigue wear tests have shown that their fatigue wear resistance performance is superior to that of equivalent steel components [8]. [9] compared to the mechanical properties of polyester ropes made from two different fibre materials, designed an accelerated creep testing method - the stepwise isothermal method, to simulate the creep behaviour of polyester ropes under long-term load conditions, and the fatigue wear of the ropes was analysed [10]. The rain flow counting method has been adopted to calculate the fatigue damage of polyester hybrid mooring lines based on the mooring tension time series. Moreover, [11] a fully coupled numerical model was established to analyse the extension behaviour of polyester mooring lines for a semi-submersible platform in the South China Sea, and the mooring response was simulated under extreme conditions to analyse the fatigue damage of the polyester mooring system. In mooring systems where the depth is less than 100 m, nylon fibre ropes are favourable for their low modulus and low cost; however, their fatigue performance after getting wet is concerning [12]. The findings of [13] indicate that nylon ropes exhibit superiority under extremely high tension, whereas under low tension, polyester ropes have a longer service life than nylon ropes. It is generally believed that polyester ropes have better fatigue resistance than nylon ropes. However, research [14] has demonstrated that significant improvements in the fatigue resistance of nylon ropes have been achieved by changing the rope twist and improving the fibre coating. In [15], the stiffness and fatigue characteristics of polyester and nylon mooring systems in floating wind turbines and found that long-twisted nylon ropes exhibited excellent fatigue performance. As the load reduces, the service life of nylon ropes is longer than that of polyester ropes, while their ultimate load also decreases, by approximately 30%. The wear resistance test

results using nylon for mooring the floating platform are poorly understood [16]. Thus, further research on the mechanical testing of polyamide ropes is required.

Owing to the unique nature of synthetic fibre materials, [17] proposed a method that combines Schapery theory with Owen's one-dimensional rheological model to describe the viscoelastic and viscoplastic behaviour of aramid and polyester synthetic fibre ropes used in deepwater mooring systems. The stiffness characteristics of the cable were no longer constant and varied changes in the load force magnitude, duration, number of cycles, etc. [18]. To study the mechanical stiffness performance, [19] explored the impact of different components of synthetic fibre ropes (including monofilaments, yarns, and braided ropes) on the stiffness of the ropes at maximum load levels and under various load paths [20]. Compared with the stiffness characteristics of high modulus polyethylene (HMPE) and aramid ropes in tensile tests, it was found that the initial stiffness of HMPE ropes increases with load owing to the rearrangement of molecules and structural adjustments (i.e., the so-called "running-in process"). However, the response of aramid ropes to this phenomenon is insignificant, as they exhibit a higher stiffness from the outset [21]. The study explored the nonlinear mechanical behaviour of polyester, aramid, and HMPE synthetic fibre ropes under cyclic loading, finding that the average load is the main factor affecting dynamic stiffness. In [22], mechanical tests were conducted on polyester and HMPE, and empirical equations were proposed considering the average tension, tension amplitude, and number of loading cycles, increasing the prediction accuracy of cable stiffness more accurate. In [23], the Kalman filter was applied to identify the parameters of empirical formulas, and it was found that the Kalman filter could provide reliable and accurate estimates for the parameters of empirical expressions. In [24], innovative strain measurement technology based on image analysis was adopted to focus on tensile tests of full-scale ropes composed of three materials: polyester, aramid, and HMPE. The study evaluated the specific impact of average load, tension amplitude, and loading frequency on the stiffness of these materials. In [25], an innovative stress-strain constitutive model was constructed based on Schapery theory and Owen's rheological theory by considering the loading history and time-dependent characteristics of synthetic fibre ropes under cyclic loading. The accuracy of the model was confirmed by comparing the predicted results of the model with actual measurement data (including dynamic stiffness and hysteresis loop data of aramid and polyester ropes under cyclic loading). In [26], a new method was developed for calculating the dynamic stiffness of ropes, utilizing the autonomous learning ability of neural networks to predict the dynamic stiffness of polyester fibre ropes. Reference [27] derived an empirical formula to predict the residual strength of polyester ropes and proposed a model using the upper and lower bound method to assess the residual strength of polyester mooring ropes. Scholars have conducted extensive stiffness test studies on synthetic fibre materials, such as nylon, polyester, aramid, and high-strength polyethylene, delving into the mechanical behaviour of these fibre ropes to ensure that these fibre ropes can be used safely and reliably in deep-sea mooring engineering [28].

This study aims to gain a deeper understanding of the mechanical properties, such as the wear resistance and quasi-static stiffness, of nylon, polyester, and high-strength polyethylene ropes, providing a theoretical basis for selecting and designing mooring anchor chains for FOWTs. First, we conducted yarn-to-yarn wear tests to evaluate the influence of different materials, twist levels, and dry/wet conditions on yarn wear failure, analysing the failure mechanisms. Subsequently, we conducted quasi-static stiffness tests on nylon and polyester ropes to study the quasi-static stiffness (i.e., their ability to resist bending and deformation) of preloaded test samples in two states: initial installation and ageing.

The rest of this paper is organized as follows: Section 2 explains yarn-to-yarn wear test fundamentals. Section 3 presents an analysis of the yarn-to-yarn wear test results. Section 4

explains fibre rope quasi-static stiffness test fundamentals. Section 5 discusses the results of the fibre rope quasi-static stiffness test. Finally, Section 6 concludes the study.

## 2. Yarn-to-yarn wear test fundamentals

### 2.1. Wear mechanism

The primary cause of wear in rope yarns is friction between the yarns. The stress on the rope unit in a static environment results from three orthogonal forces acting at the centroid of the cross-section and the moments acting along each coordinate axis, as shown in Fig 1. Although the work generated by each local slip is minimal, the accumulated frictional force can become significant when a slip occurs at multiple positions within the rope.

When the rope is under tension, contact occurs between the yarns. Relative parallel sliding occurs in the direction of the contact lines between yarns in the same layer. This contact force and the relative sliding between continuous helical yarns result in frictional forces opposite to the direction of the relative motion. Fig 2(A) illustrates two adjacent contacting yarn units, where $b$ represents the contact point, and $a$ and $c$ are reference lines. Throughout the process, the slip $S$ can be expressed as $S = d[(1 + \varepsilon_s)tan\theta_0 - tan\theta]$, where $d$ denotes the diameter of

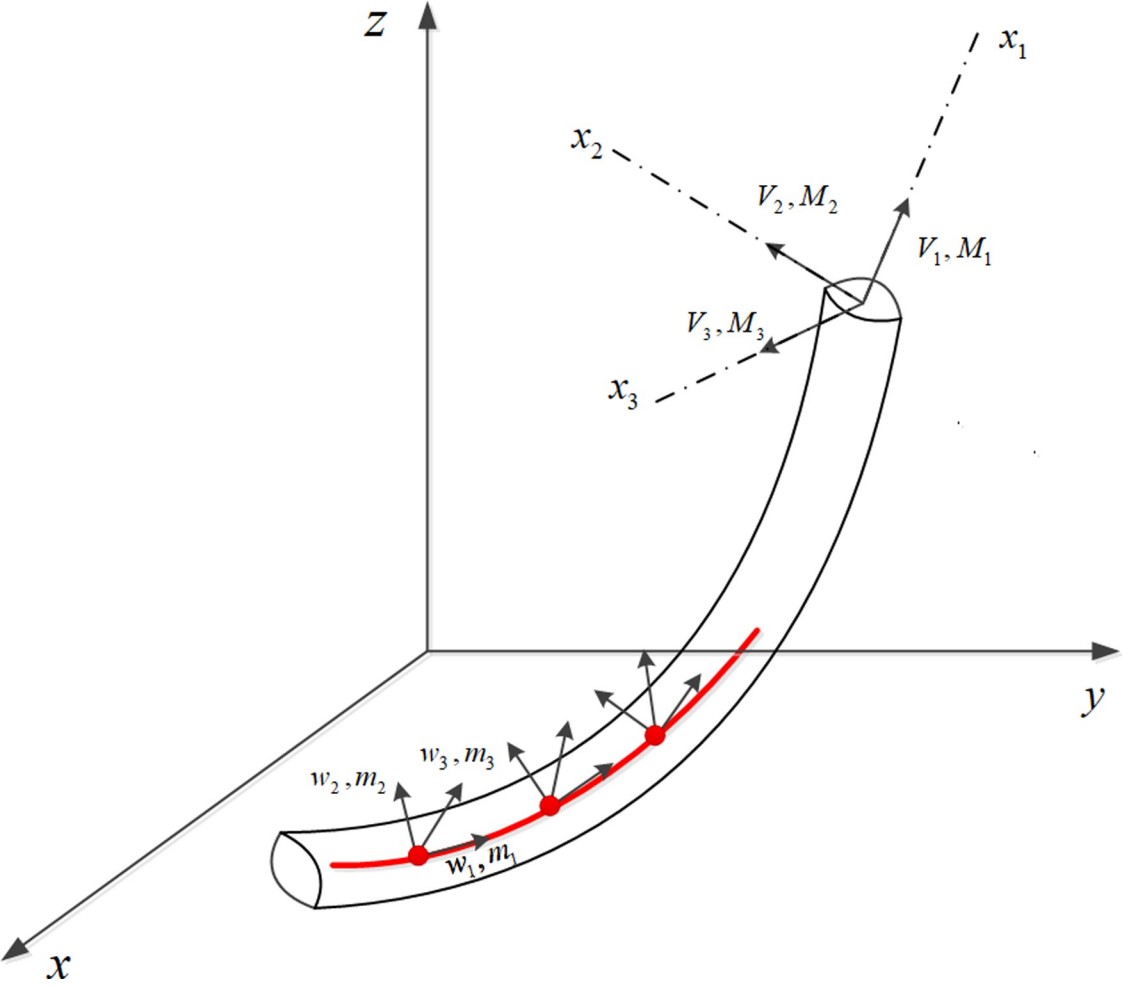

**Fig 1. Stress on a fibre unit in a static environment.**

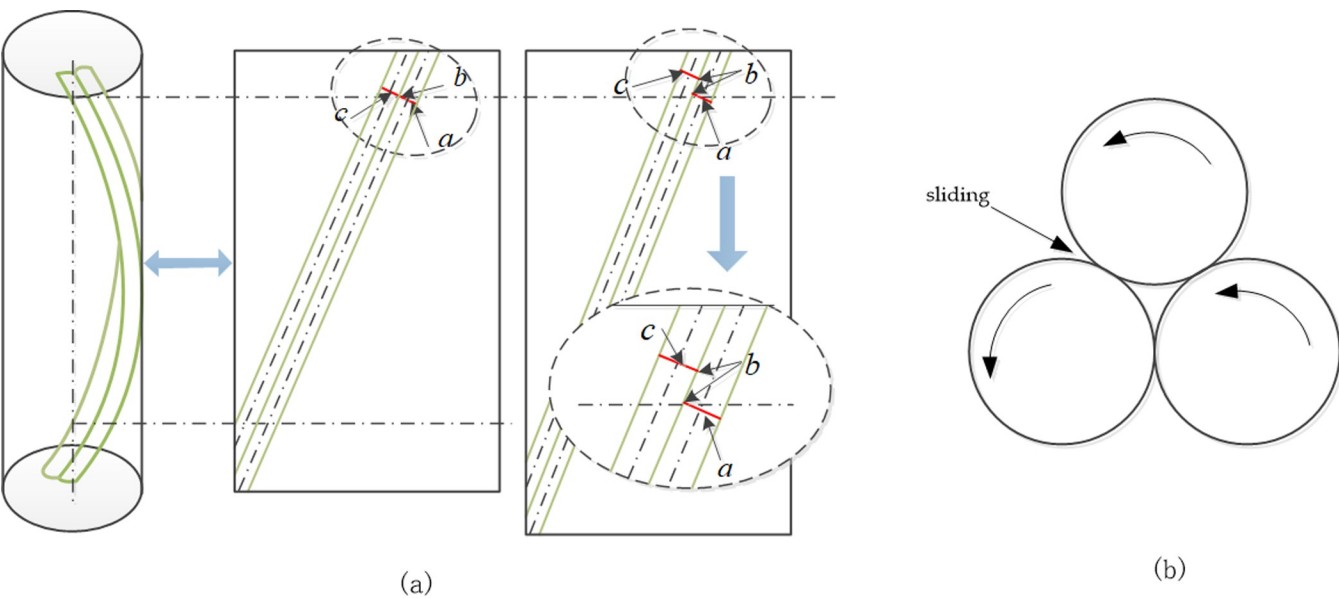

**Fig 2. Friction generated by the motion of fibre yarn units.** (a) Axial sliding; (b) Twisting sliding.

the helical yarn, $\varepsilon_s$ denotes the strain generated in the helical yarn, $\theta_0$ denotes the initial helix angle, and $\theta$ represents the helix angle after slipping. Twisting sliding friction refers to the frictional torque generated owing to the twisting tendency of rope units, as shown in Fig 2(B), wherein the magnitude is proportional to the unit length. When a unit of the rope experiences interruption or is located at the endpoint, the torque generated by twisting sliding friction prevents the rope from slackening or uncoiling.

## 2.2. Test equipment

This experiment was conducted using the assembled rope wear dynamic test machine provided by Zhejiang Four Brothers Rope Co., Ltd. The test machine primarily comprised a rope traction wheel, pulley assembly, weight hammer, and monitoring system. The details of its structure are shown in Fig 3. The test was conducted following the requirements of the Cordage Institute and ASTM D 6611 (2007) [29] to assess the wear between yarns. This test is an industry standard for evaluating the wear resistance between synthetic fibres and provides a qualitative wear resistance value.

The wear test involved winding a length of yarn between the traction wheel and the pulley assembly. Three pulleys in the assembly were geometrically arranged to produce a specific apex angle, and the apexes of the yarn were twisted and interwoven. A weight hammer was attached to one end of the yarn to apply constant tension. At the other end, a motor drove the rope back and forth at a frequency of 0.83 Hz until the yarn failed owing to self-wear in the twisted wrapping area. A diagram of the test machine is shown in Fig 4.

In Fig 4, $H$ represents half the distance between the centres of the two pulleys, $V$ denotes the perpendicular distance from the centre of the lower pulley to the centreline of the upper pulley, $L_1$ represents the length of the yarn from the tangent of the upper pulley to the upper apex, $L_2$ represents the length of the yarn from the tangent of the lower pulley to the upper apex, $W$ represents the length of the yarn winding portion, $\beta$ represents the apex angle of the yarn, $\alpha$ represents half of the apex angle of the yarn, and $r$ represents the radius of the pulley.

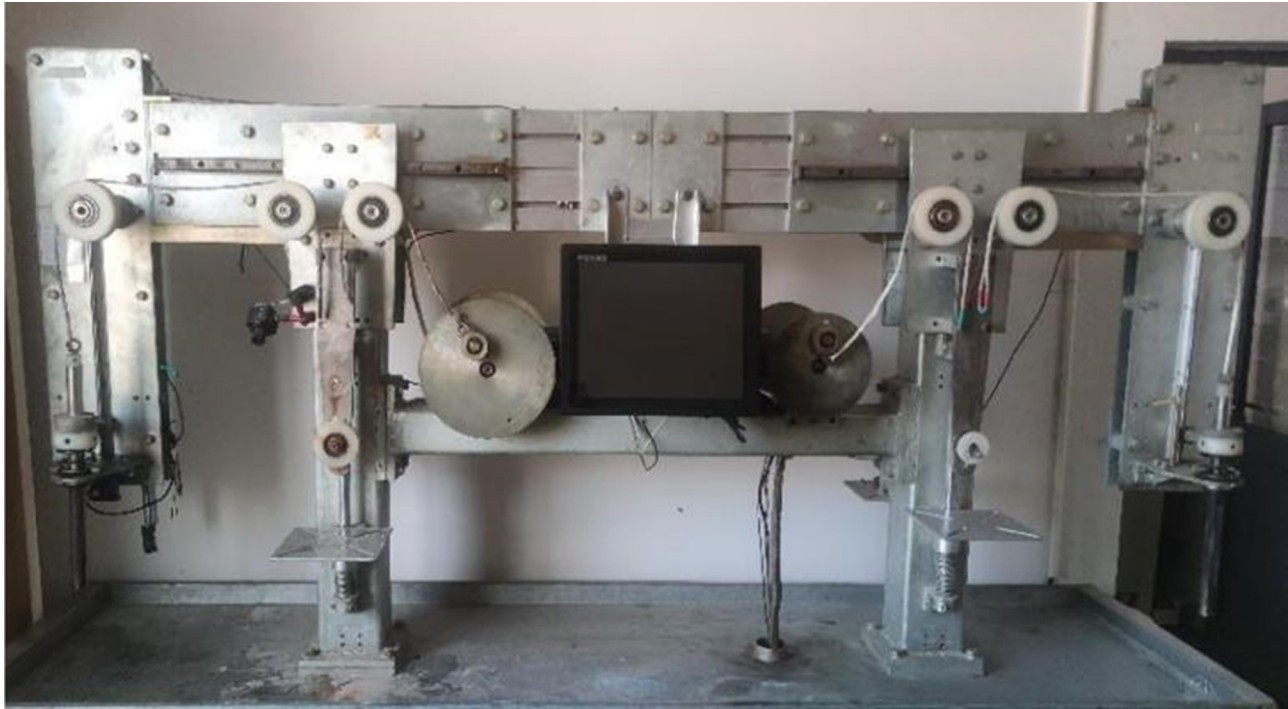

**Fig 3. Structure of the dynamic rope wear test machine.**

During yarn-on-yarn abrasion testing, it is challenging to accurately determine the yarn winding angle because it is difficult to detect angle errors. Generally, a measurement error range of 4˚–6˚ occurs, and it is extremely rare to achieve an error of 2˚. By contrast, measuring the distance between the centres of the pulleys is considerably simpler. Eqs (1) to (3) provide a method for calculating the yarn angle based on the distance between the centres of the pulleys. By applying these equations to determine the angle of the yarn vertices, rather than direct measurement, the accuracy of the measurement can be significantly improved. In the calculation process, it is assumed that all pulley radii are the same and that the yarn is wound around the centre of the lower pulley on the vertical line, with the two vertex angles of the yarn being equal. The calculation process is as follows:

$$V = L_1 \cos\alpha + L_2 \cos\alpha + W \Rightarrow \cos\alpha = \frac{V - W}{L_1 + L_2}, \tag{1}$$

$$\begin{array}{l} L_2 \sin\alpha = r\cos\alpha \\ H = L_1 \sin\alpha + r\cos\alpha \end{array} \Rightarrow \sin\alpha = \frac{H}{L_1 + L_2}, \tag{2}$$

$$\tan\alpha = \frac{\sin\alpha}{\cos\alpha} = \frac{H}{V - W} \Rightarrow \beta = 2\arctan\left(\frac{H}{V - W}\right). \tag{3}$$

## 2.3. Operating condition design

The experiment was conducted in an environment with a constant room temperature stabilised at $21 \pm 3$˚C and a humidity maintained in the 40–60% range. The load was applied

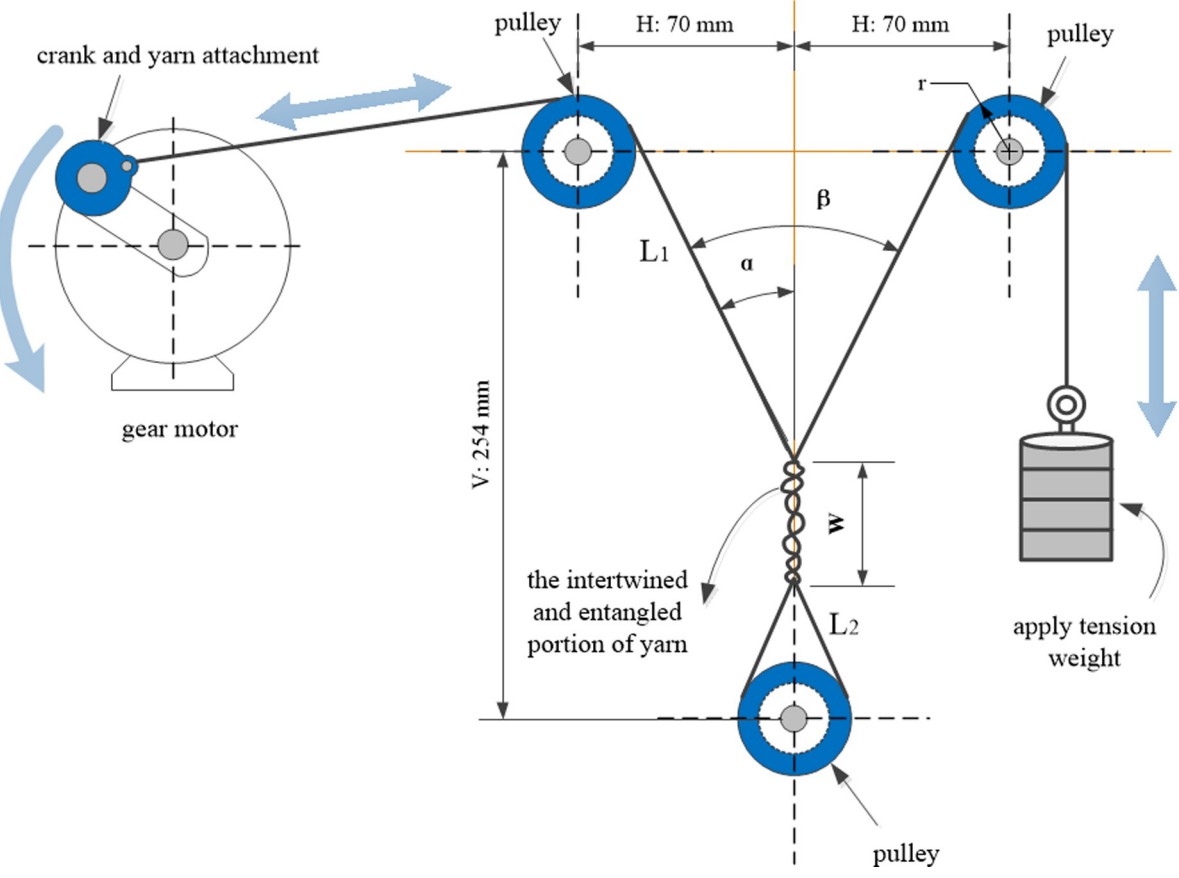

**Fig 4. Diagram of the structure of the yarn wear test machine.**

vertically, and the value of the vertical load was determined based on the strength load of the yarn. The gear motor speed was adjusted until the yarn movement reached a speed of 50 r/min; that is, the motor caused the rope yarn to undergo reciprocal friction at a frequency of 0.83 Hz until the yarn failed owing to self-wear in the twisted wrapping area. The experimental conditions are listed in Table 1.

## 3. Analysis of yarn-to-yarn wear test results

### 3.1. Influence of twist level on yarn

The impact of the yarn twist level on wear resistance varied across different fibre types. Figs 5 and 6 show that there was no significant difference in wear and breakage between PET fibres with twist levels of 5 and 10. Thus, it is unclear whether a low or high twist is more wear-resistant. Fig 7 demonstrates that for HMPE fibre yarns, a twist level of ten exhibited higher wear resistance compared to a twist level of five. However, for PA fibres, the wear resistance of twist level five exceeded that of twist level ten, as depicted in Fig 8.

HMPE is currently recognised as a synthetic fibre material with high strength and high tensile properties. Compared with PET and PA fibres, HMPE is less sensitive to changes in twist. For PET and PA fibres, it is necessary to not have an exceedingly high twist because an excessive twist can lead to the yarn becoming extremely hard and brittle, thereby reducing its softness and bending properties. Additionally, an excessively high twist can also cause mutual

**Table 1. Fibre yarn abrasion test conditions.**

| Condition | Suppliers | Twists (turns/ m) | Absolute load (g) | Strength load (mN/tex) | Frequency (times/min) | Dry/wet conditions | Cycle count | Breakage situation description | Description |
|---|---|---|---|---|---|---|---|---|---|
| | | | | **Polyester---Polyethylene Terephthalate (PET) fibre** **Yarn density: 3000D-3033D** | | | | | |
| 1 | GuXian Dao | 5 | 1299 | 39 | 50 | dry | 6336 | crossing point | with marine lubricant additive |
| 2 | GuXian Dao | 5 | 1299 | 39 | 50 | dry | 7786 | crossing point | |
| 3 | GuXian Dao | 5 | 1299 | 39 | 50 | dry | 7541 | crossing point | |
| 4 | GuXian Dao | 10 | 1299 | 39 | 50 | dry | 7850 | crossing point | |
| 5 | GuXian Dao | 10 | 1299 | 39 | 50 | dry | 7140 | crossing point | |
| 6 | GuXian Dao | 10 | 1299 | 39 | 50 | dry | 8898 | crossing point | |
| 7 | GuXian Dao | 5 | 1299 | 39 | 50 | wet | 42527 | crossing point | |
| 8 | GuXian Dao | 5 | 1299 | 39 | 50 | wet | 29228 | crossing point | |
| 9 | GuXian Dao | 5 | 1299 | 39 | 50 | wet | 25638 | crossing point | |
| 10 | GuXian Dao | 10 | 1299 | 39 | 50 | wet | 25317 | crossing point | |
| 11 | GuXian Dao | 10 | 1299 | 39 | 50 | wet | 23928 | crossing point | |
| 12 | GuXian Dao | 10 | 1299 | 39 | 50 | wet | 25856 | crossing point | |
| 13 | GuXian Dao | 5 | 1299 | 39 | 50 | dry | 9239 | crossing point | without marine lubricant additive |
| 14 | GuXian Dao | 5 | 1299 | 39 | 50 | dry | 6508 | crossing point | |
| 15 | GuXian Dao | 5 | 1299 | 39 | 50 | dry | 7284 | crossing point | |
| 16 | GuXian Dao | 10 | 1299 | 39 | 50 | dry | 9887 | crossing point | |
| 17 | GuXian Dao | 10 | 1299 | 39 | 50 | dry | 6019 | crossing point | |
| 18* | GuXian Dao | 10 | 1299 | 39 | 50 | dry | 2418 | knotting point | |
| 19 | GuXian Dao | 5 | 1299 | 39 | 50 | dry | 28261 | crossing point | |
| 20 | GuXian Dao | 5 | 1299 | 39 | 50 | wet | 34672 | crossing point | |
| 21 | GuXian Dao | 5 | 1299 | 39 | 50 | wet | 41497 | crossing point | |
| 22 | GuXian Dao | 10 | 1299 | 39 | 50 | wet | 27040 | crossing point | |
| 23 | GuXian Dao | 10 | 1299 | 39 | 50 | wet | 27414 | crossing point | |
| 24 | GuXian Dao | 10 | 1299 | 39 | 50 | wet | 21036 | crossing point | |
| | | | | **Dyneema---HMPE fibre** **Yarn density: 1600D** | | | | | |

*(Continued)*

**Table 1.** (Continued)

| Condition | Suppliers | Twists (T/m) | Absolute load (g) | Strength load (mN/tex) | Frequency (times/min) | Dry/wet conditions | Cycle count | Breakage situation description | Description |
|---|---|---|---|---|---|---|---|---|---|
| 1 | DuPont | 5 | 1120 | 62 | 50 | dry | 22 | crossing point | |
| 2 | DuPont | 5 | 1120 | 62 | 50 | dry | 21 | crossing point | |
| 3 | DuPont | 5 | 1120 | 62 | 50 | dry | 20 | crossing point | |
| 4 | DuPont | 10 | 1120 | 62 | 50 | dry | 106 | crossing point | |
| 5 | DuPont | 10 | 1120 | 62 | 50 | dry | 127 | crossing point | |
| 6 | DuPont | 10 | 1120 | 62 | 50 | dry | 131 | crossing point | |
| 7 | DuPont | 5 | 1120 | 62 | 50 | wet | 1926 | crossing point | |
| 8 | DuPont | 5 | 1120 | 62 | 50 | wet | 2378 | crossing point | |
| 9 | DuPont | 5 | 1120 | 62 | 50 | wet | 2846 | crossing point | |
| 10 | DuPont | 10 | 1120 | 62 | 50 | wet | 3627 | crossing point | |
| 11 | DuPont | 10 | 1120 | 62 | 50 | wet | 2602 | crossing point | |
| 12 | DuPont | 10 | 1120 | 62 | 50 | wet | 3294 | crossing point | |

**Nylon---Polyamide (PA) fibre**
**Yarn density: 1800D×2**

| Condition | Suppliers | Twists (T/m) | Absolute load (g) | Strength load (mN/tex) | Frequency (times/min) | Dry/wet conditions | Cycle count | Breakage situation description | Description |
|---|---|---|---|---|---|---|---|---|---|
| 1 | DK | 5 | 940 | 44 | 50 | dry | 85118 | crossing point | |
| 2 | DK | 5 | 940 | 44 | 50 | dry | 61017 | crossing point | |
| 3 | DK | 5 | 940 | 44 | 50 | dry | 85437 | crossing point | |
| 4 | DK | 10 | 940 | 44 | 50 | dry | 32399 | crossing point | |
| 5 | DK | 10 | 940 | 44 | 50 | dry | 26491 | crossing point | |
| 6 | DK | 10 | 940 | 44 | 50 | dry | 30198 | crossing point | |
| 7 | DK | 5 | 940 | 44 | 50 | wet | 1347 | crossing point | |
| 8 | DK | 5 | 940 | 44 | 50 | wet | 1703 | crossing point | |
| 9 | DK | 5 | 940 | 44 | 50 | wet | 1541 | crossing point | |
| 10 | DK | 10 | 940 | 44 | 50 | wet | 440 | crossing point | |
| 11 | DK | 10 | 940 | 44 | 50 | wet | 459 | crossing point | |
| 12 | DK | 10 | 940 | 44 | 50 | wet | 364 | crossing point | |

*Note: Owing to an error in the connection between the test sample and the gear motor, the PET fibre encountered problems during the test at condition 18. Specifically, the yarn broke at the fixed knotting point, resulting in a measured cycle count of only 2418. Subsequently, we conducted additional tests for this condition to ensure the accuracy of the data. The test results found that the cycle count at which the yarn broke at the crossing point significantly increased to 7368. In the subsequent analysis of the test results, both sets of data will be presented, corresponding to Figs 6(A), 9(B) and 11(A), respectively.

damage to the fibres within the yarn, affecting the abrasion resistance of the yarn. However, for HMPE material, its fibres can withstand greater strength loads, and the impact of twist on the internal structure of the yarn is smaller, and the higher the twist, the better the wear resistance. Therefore, when choosing the twist, it is necessary to comprehensively consider factors such as the mechanical properties of the yarn material and fibres to achieve the best durability and wear resistance.

### 3.2. Influence of dry and wet conditions on yarn wear

In the wear tests for the PET and HMPE yarns, the wear resistance was varied under dry and wet conditions. As observed in Figs 9 and 10, wet yarns exhibited greater wear resistance. This was because the lubricating effect of water on the fibres reduced friction and thus decreased yarn wear. Additionally, moisture filled the gaps within the yarn, increasing its density,

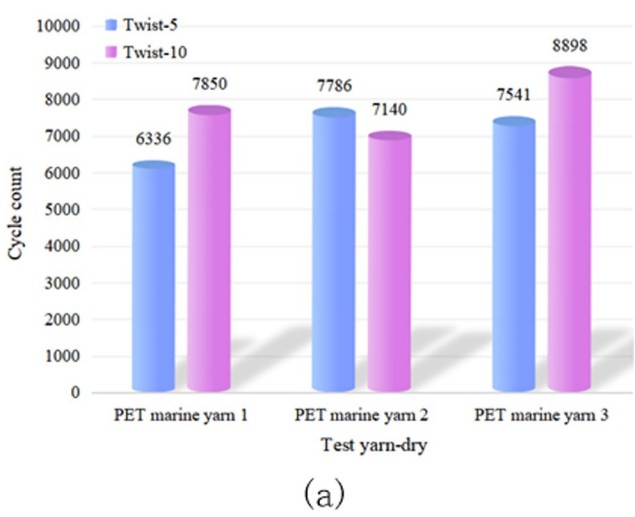
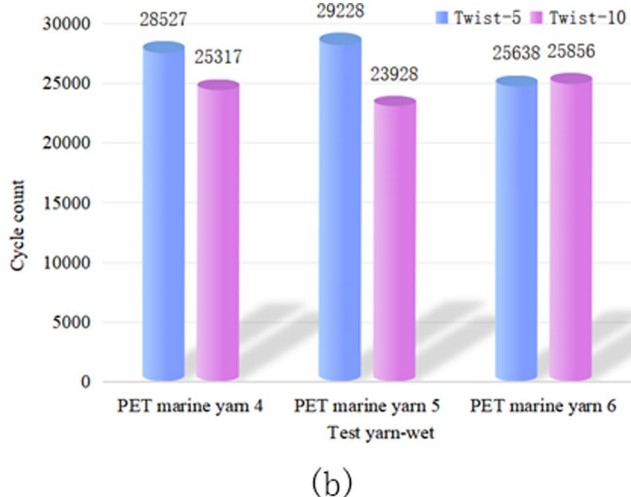

**Fig 5.** Influence of twist level on the wear fracture of PET yarns with marine oil agents in (a) dry and (b) wet conditions.

toughness, and wear resistance. Conversely, the yarns in dry environments experienced continuous friction, which could potentially melt the fibres owing to localised high temperatures.

However, it must be noted that nylon has its unique properties. As illustrated in Fig 10(B), the wear resistance of nylon fibres significantly decreases when wet owing to the presence of polar hydrophilic groups in its molecular structure—specifically polar amide groups—which can form hydrogen bonds with water molecules in the environment. This interaction reduces the regularity of the molecular chains and decreases the number of hydrogen bonds between them, thereby weakening the intermolecular forces. Consequently, the wet state has a substantial impact on the stability and mechanical properties of nylon fibres.

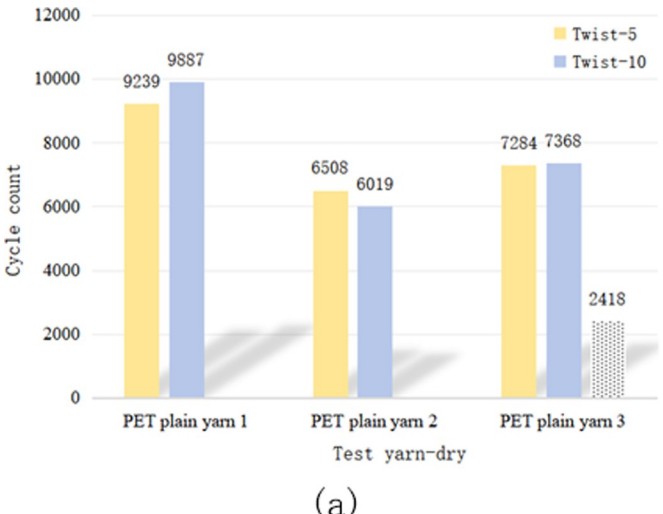
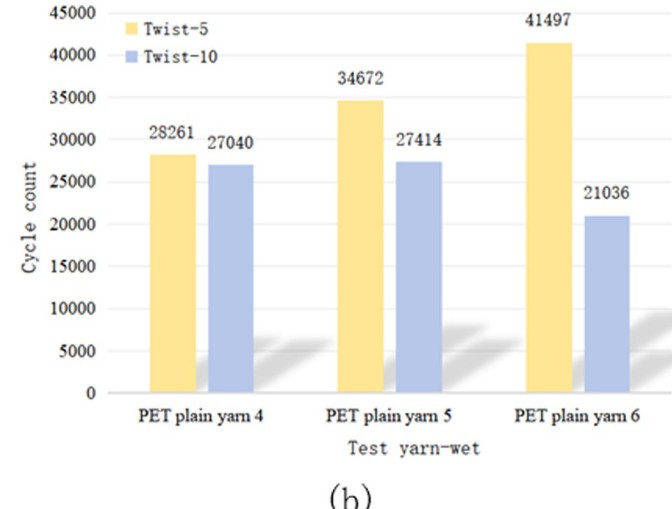

**Fig 6.** Influence of twist level on the wear fracture of PET yarns without marine oil agents in (a) dry and (b) wet conditions.

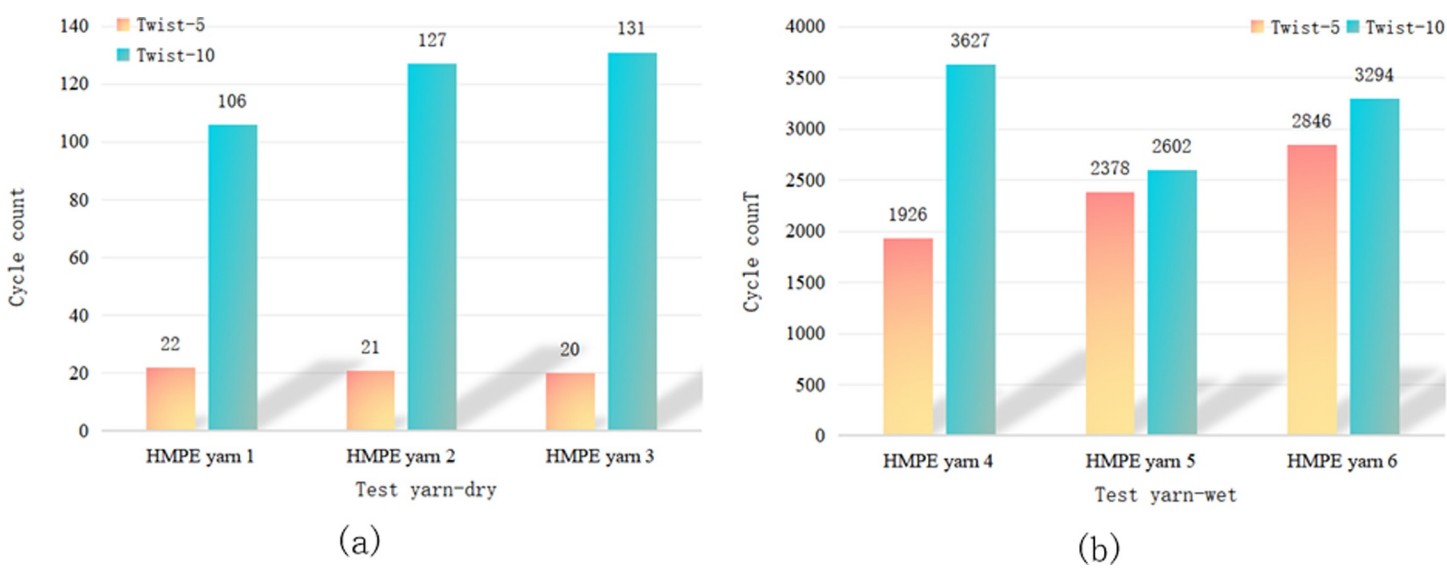

**Fig 7.** Influence of twist level on the wear fracture of HMPE yarns in (a) dry and (b) wet conditions.

## 3.3. Impact of the fibre material on yarn wear

Generally, ropes in marine environments are coated with a layer of protective oil to shield them from various corrosive factors, such as chloride ions, salt, and ultraviolet rays, thereby enhancing their durability. For this study, a comparative analysis of the wear resistance was conducted between marine ropes coated with marine oil and ordinary ropes without oil coating. The marine oil coating material selected was polytetrafluoroethylene (PTFE), a material with excellent resistance to high temperatures and corrosion, commonly used on the surface of fibre ropes to enhance their wear resistance and durability.

From the experimental results in Fig 11, it can be observed that under dry conditions, the wear resistance of marine yarns and ordinary yarns is not significantly different. This may be

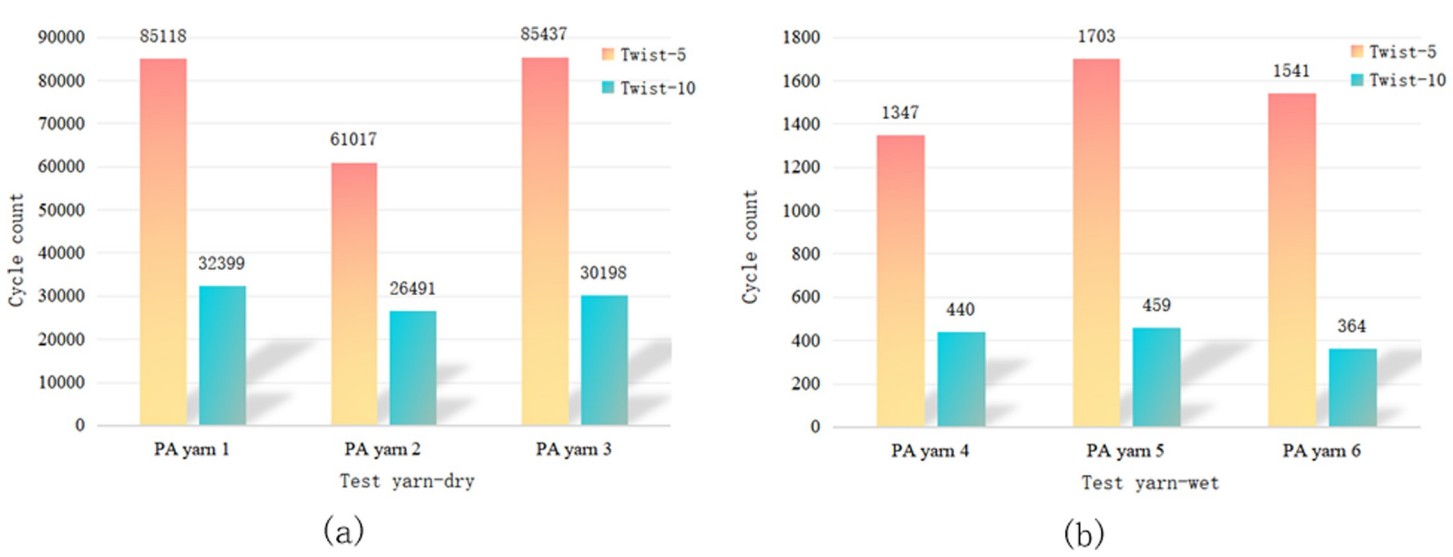

**Fig 8.** Influence of twist level on the wear fracture of PA yarns in (a) dry and (b) wet conditions.

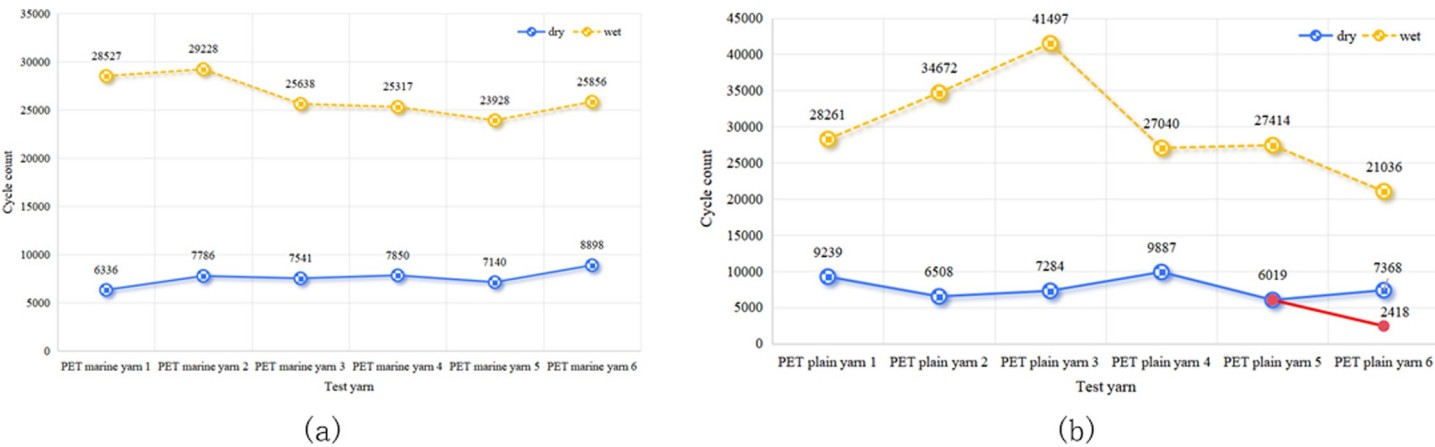

**Fig 9.** Impact of dry and wet conditions on the wear fracture of PET yarns (a) with and (b) without marine oil agents.

because the protective effect of the marine oil agent could not be fully utilised in the absence of a moist environment, as illustrated in Fig 11(A). However, when the tests were conducted in a moist environment, the results changed, as illustrated in Fig 11(B). The marine PET yarns coated with the marine oil agent exhibited a longer break cycle under moist conditions, indicating higher wear resistance. This suggests that the marine oil agent can more effectively exert its protective function in a moist environment, thereby extending the service life of the ropes and reducing wear damage. Therefore, by applying these high-performance materials for coating treatment, fibre ropes can better withstand various external factors in the marine environment, extend their service life, and ensure their safe and reliable use.

Under dry conditions, no significant difference was observed in the wear resistance between the marine and ordinary ropes, possibly because the protective effect of the marine oil could not be fully realised in the absence of a moist environment, as indicated in Fig 11(A). However, the results differed when the test environment was moist, as depicted in Fig 11(B). The PET marine ropes coated with marine oil exhibited a greater number of fracture cycles, indicating higher wear resistance in moist conditions. This suggests that marine oil is more

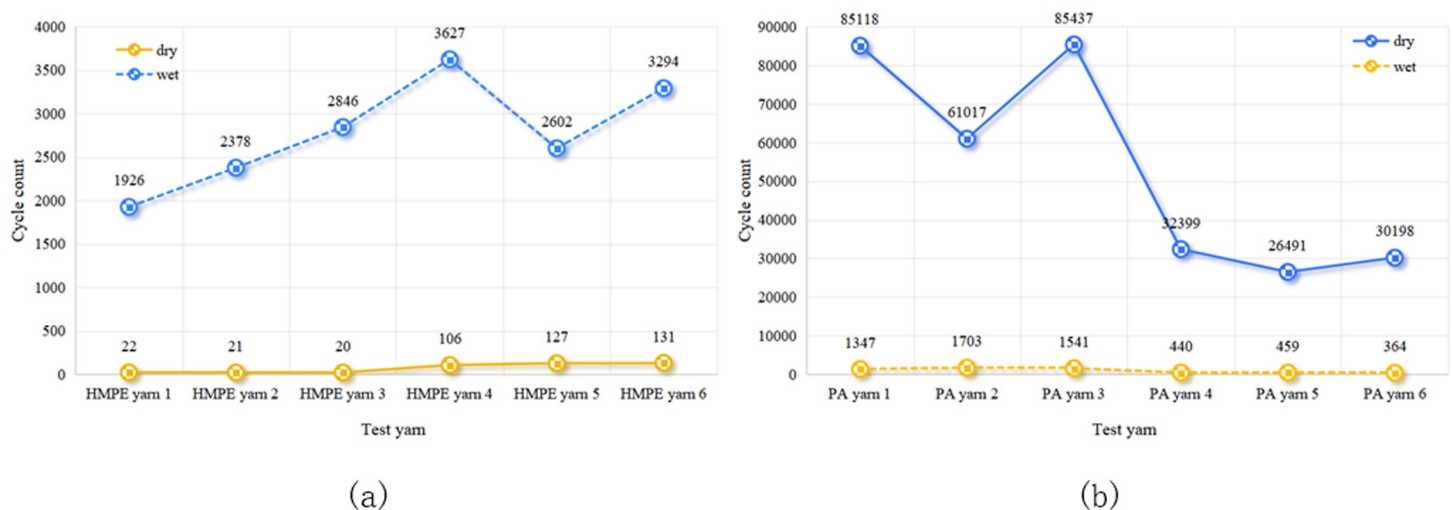

**Fig 10.** Effect of dry and wet conditions on the abrasion and breakage of fibre yarns (a) HMPE (b) PA.

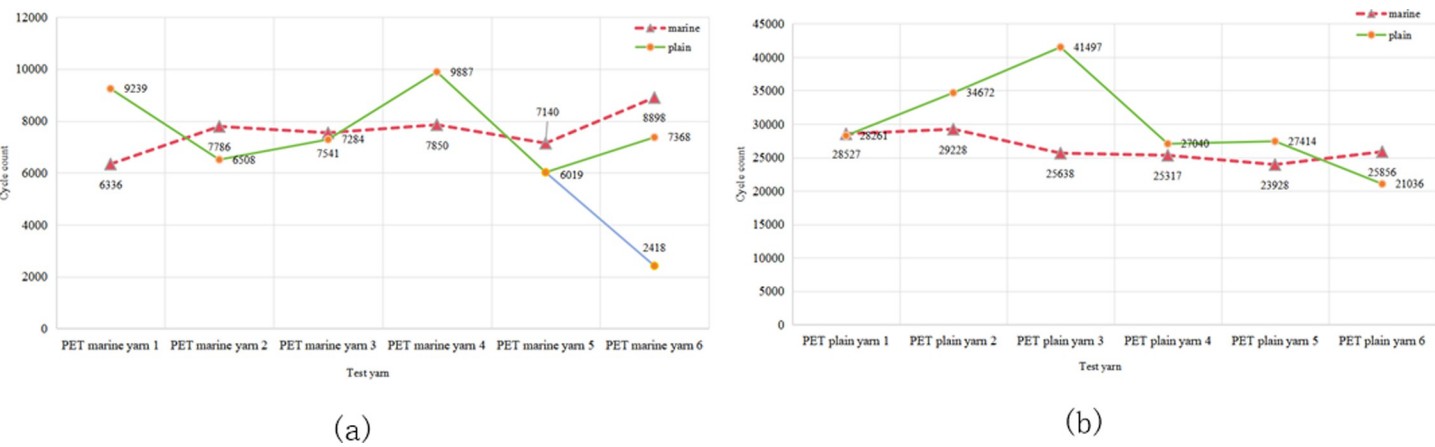

**Fig 11.** Impact of marine oil agents on the wear fracture of PET yarns under (a) wet and (b) dry conditions.

effective in moist environments, as it can prolong the service life of ropes and reduce damage caused by wear.

Fig 12 presents a comparison between the wear conditions of different materials, indicating that the friction cycles of HMPE yarns were significantly lower than those of PET yarns. The reasons for this difference may be multifaceted. One possibility is that the lower yarn density and relatively thinner yarns of HMPE samples cause them to be more susceptible to wear under the action of friction. Additionally, heat accumulation generated by friction in dry environments may lead to localised fibre melting. Compared to HMPE, PET fibres exhibit superior heat resistance. These factors collectively contributed to the relatively inferior wear performance of HMPE yarns observed in the experiment.

Furthermore, as observed in Fig 12, PA exhibits strong wear resistance. This may be attributed to the crystalline nature of PA molecules, which restricts the movement of molecular chains and enhances their strength. However, the wear resistance of PA fibres sharply declined in moist conditions, as shown in Fig 12(B), highlighting the need for careful consideration. Dry and wet conditions significantly affected the stability and mechanical properties of PA

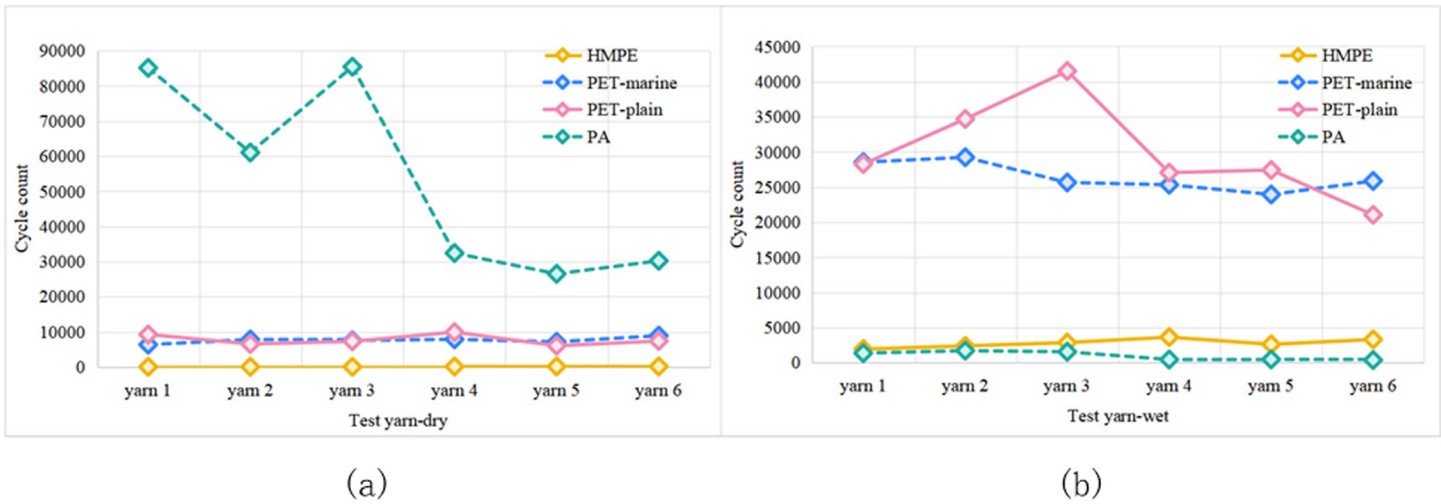

**Fig 12.** Influence of different fibre materials on the wear fracture of yarns in (a) dry and (b) moist conditions.

fibres. Although the wear resistance of PA fibres decreased in moist conditions, PA fibres with adequate water-resistant coating treatment can still be an excellent choice for mooring ropes in marine environments.

## 4. Quasi-static stiffness test fundamentals

### 4.1. Stiffness calculation method

The expression for rope stiffness is derived based on modelling experiments to identify the various physical quantities that affect stiffness; it is expressed as

$$K_r = EA/MBS = f(MBS, T_m, T_A, N, D, T, L, d, \varepsilon, \rho, \ldots), \tag{4}$$

where $EA/MBS$ denotes the dynamic stiffness, $MBS$ denotes the minimum breaking strength, $T_m$ denotes the average tension, $T_A$ denotes the tension amplitude, $N$ indicates the number of cycles, $D$ denotes the rope diameter, $T$ denotes the load cycle, $L$ denotes the initial rope length, $\rho$ denotes the rope density, $\varepsilon$ denotes the strain, and $d$ denotes the line density.

Select $D, d, T$ as the fundamental dimensions, and combine with $\pi$ theorem to obtain the similarity dimensionless numbers [22]:

$$\begin{cases} \pi_1 = \dfrac{E/\rho}{MBS/d} \\ \pi_2 = D \\ \pi_3 = \dfrac{MBS \times T^2}{\rho \times l^2} \\ \pi_4 = \dfrac{T_m \times T^2}{\rho \times l^2} \\ \pi_5 = \varepsilon, \pi_6 = N \end{cases} \tag{5}$$

Finally, dynamic stiffness $K_r$ can be represented as:

$$K_r = f_1(\pi_1, \pi_2, \pi_3, \pi_4, \pi_5, \pi_6) \tag{6}$$

According to the similarity numbers in formulas (5) and (6), and combining the similarity number of dynamic modulus between the model and prototype first proposed by Fernandes and Rossi [30], the similarity number for dynamic stiffness of ropes made of the same material and with the same structure can be obtained:

$$\lambda_{EA/MBS} = \frac{(EA/MBS)_m}{(EA/MBS)_P} = \frac{\lambda_{E/\rho}}{\lambda_{MBS}} = 1 \tag{7}$$

According to Eq (7), the stiffness patterns of large-scale ropes can be investigated using model experiments on small-scale ropes, saving time and experimental costs. Considering the time-related characteristics of fibre ropes in mooring analyses. Moreover, the American Bureau of Shipping (ABS) also specifies the equation for static stiffness as

$$K_{rs} = (T_2 - T_1)/[\varepsilon_2 - \varepsilon_1 + Clg(t)], \tag{8}$$

where $K_{rs}$ is the static stiffness coefficient, $T_1$ denotes the initial tension, $T_2$ denotes the final test tension, $\varepsilon_1$ denotes the initial strain, $\varepsilon_2$ denotes the final strain, $C$ denotes the creep coefficient, and $t$ denotes the creep time.

## 4.2. Test equipment

This experiment was conducted using a 3000-t microcomputer-controlled horizontal heavy-duty tension testing machine from Zhejiang Four Brothers Rope Co., Ltd., primarily comprising the equipment base, stator/stator hydraulic device, test end/test hydraulic device, and water environment system, as shown in Fig 13. The test end was equipped with four tension rods that pulled the rope in reverse via a reciprocating motion. Simultaneously, the displacement sensor at the free end continuously recorded the elongation rate $\Delta L$ of the rope every 0.3 s for each test condition.

A fibre rope braided based on a $1 \times 12$ configuration was selected as the test specimen, with the main parameters listed in Table 2. The specimen was fixed to the testing equipment using the eye-splice method, with a total length of 14 m. The eye-splice insertion length of the sample was approximately 3 m, and the middle 5 m section was chosen as the experimental measurement length, as illustrated in the diagram of the test rope shown in Fig 14.

The samples were immersed in water for at least 4 h before testing. Subsequently, according to the sample size, the extensometer and measuring gauge were installed, and the measuring clamp of the displacement meter was fixed at the marked section to ensure that the sensor recorded the change in displacement of the rope test measurement section, as shown in Fig 15.

## 4.3. Preloading test

A preloading test was conducted to ensure that each fibre inside the rope was under equal tension and to stabilise the internal structure of the rope. The sample rope was installed between the two jaws of the tension testing machine, and an extensometer and measuring gauge were installed on the specimen to record the elongation of the rope. A preload of 1% of the minimum breaking strength (MBS) was applied and maintained for 5 min. Thereafter, the loads on both the nylon and polyester ropes were increased to 3.6% MBS and 13% MBS, respectively, and maintained for 2 h. Subsequently, the loads on both the nylon and polyester ropes were further increased to 20% MBS and 40% MBS, respectively, and maintained for 3 h. Finally, the rope loads were reduced back to 3.6% MBS and 13% MBS and maintained for 6 h to complete the preloading test.

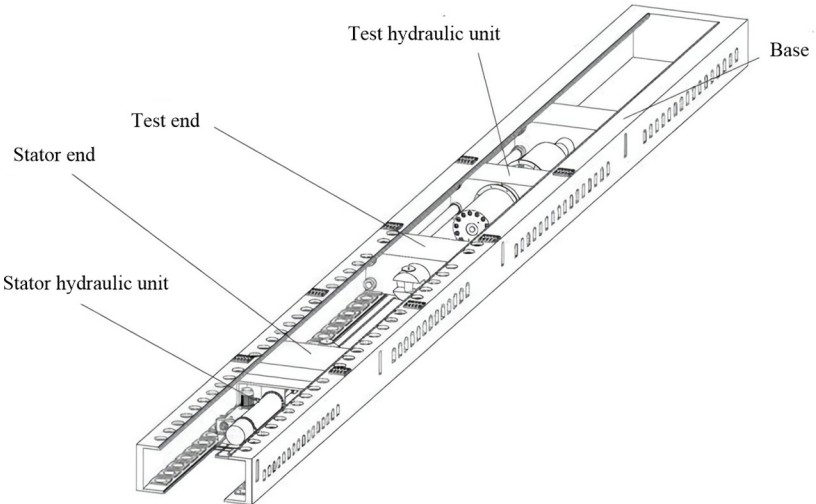

**Fig 13. Structure diagram of a 3000 tons and below rope tensile reciprocating test machine.**

**Table 2. Detailed specifications of the synthetic fibre rope test samples.**

| Sample | Material | Strand | Material | Total | Minimum breaking strength |
|--------|----------|--------|----------|-------|---------------------------|
| rope 1 | PET | 1 × 12 strand | 68 mm | 14 m | 1786.4 kN |
| rope 2 | PA | 1 × 12 strand | 72 mm | 14 m | 1166.7 kN |

Fig 16 presents the load variations and the corresponding deformation of the rope during the preloading test. The strain of the nylon rope was greater than that of the polyester rope throughout the preloading phase. This indicates that nylon material is relatively soft and can undergo relatively large deformation even under low loads, whereas the strain of the polyester rope is relatively small compared to the nylon rope.

Fig 17 illustrates the stress–strain relationship for the synthetic fibre rope during the preloading phase. The figure shows that the strain of the polyester rope reached its maximum at 6.368% under a load of 40% MBS, decreasing to 4.808% when the load was reduced to 13% MBS, resulting in a rope deformation change of approximately 1.560% within a short period. Following static holding for 6 h, the strain of the polyester rope was 4.103%, corresponding to a decrease of 0.705% compared to the strain before static holding. This implies that even after the preloading process was completed, the rope was subjected to residual structural deformation, and the rope contracted during the prolonged static holding period. This phenomenon was also evident in the case of the nylon rope. The strain of this rope at the maximum load was 11.180%, decreasing to 5.633% when the load was reduced to 3.6% MBS, resulting in a change in the rope deformation of approximately 5.547% within a short period. Following static holding for 6 h, the strain of the nylon rope was 4.377%, corresponding to a decrease of 1.256% compared to the strain before static holding. A comparison between the performance of the nylon and polyester ropes revealed that the deformation of the nylon rope was larger and that its structural contraction was more pronounced.

## 5. Analysis of the rope fibre quasi-static stiffness test results

### 5.1. Quasi-static rope stiffness test during the initial installation stage

Static stiffness tests were conducted on two types of pre-treated ropes: initially installed ropes and aged ropes. The static stiffness test process for the synthetic fibre ropes involved increasing the initial load (13% MBS for polyester and 3.6% MBS for nylon) to 30% MBS and maintaining it for 100 min, then increasing the tension to 45% MBS and 60% MBS for the polyester and nylon

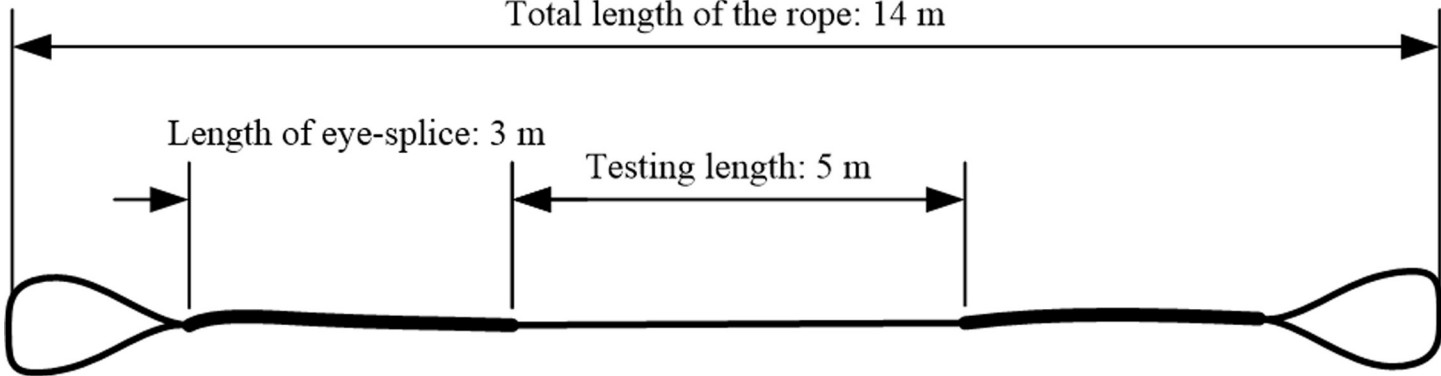

**Fig 14. Diagram of the synthetic fibre rope test segment model.**

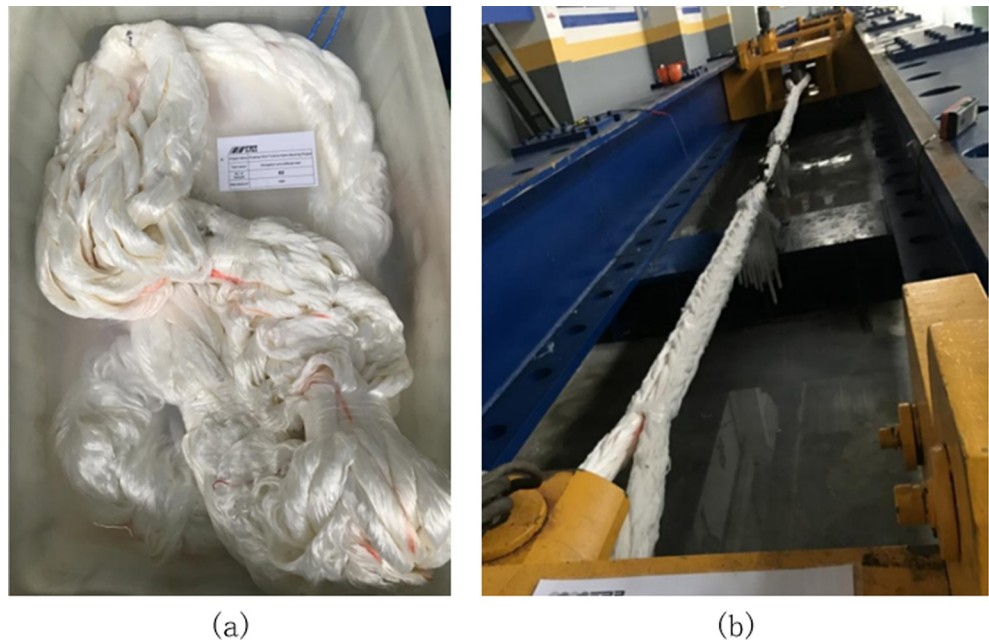

**Fig 15.** Immersion and installation of the synthetic fibre rope: (a) rope soaking and (b) load stretching.

ropes, respectively, and maintaining it for 100 min before unloading to the initial load. The rope was then kept under the initial load for at least 200 min. The results are shown in Fig 18.

The static stiffness test was designed with multiple loads: 30%, 45%, and 60% MBS. Different loads corresponded to specific creep coefficients. To obtain the creep coefficients under different loads, tests were conducted at 1, 10, and 100 min according to the requirements of the American Bureau of Shipping (ABS) regulations, and the elongation of the rope was measured and recorded. Fig 18 shows that under a constant load, the rope gradually elongated, and the elongation of the nylon rope was greater than that of the polyester rope. Both Figs 18 (B) and 19 show the strain of the rope at different points in time. Upon unloading, the strain

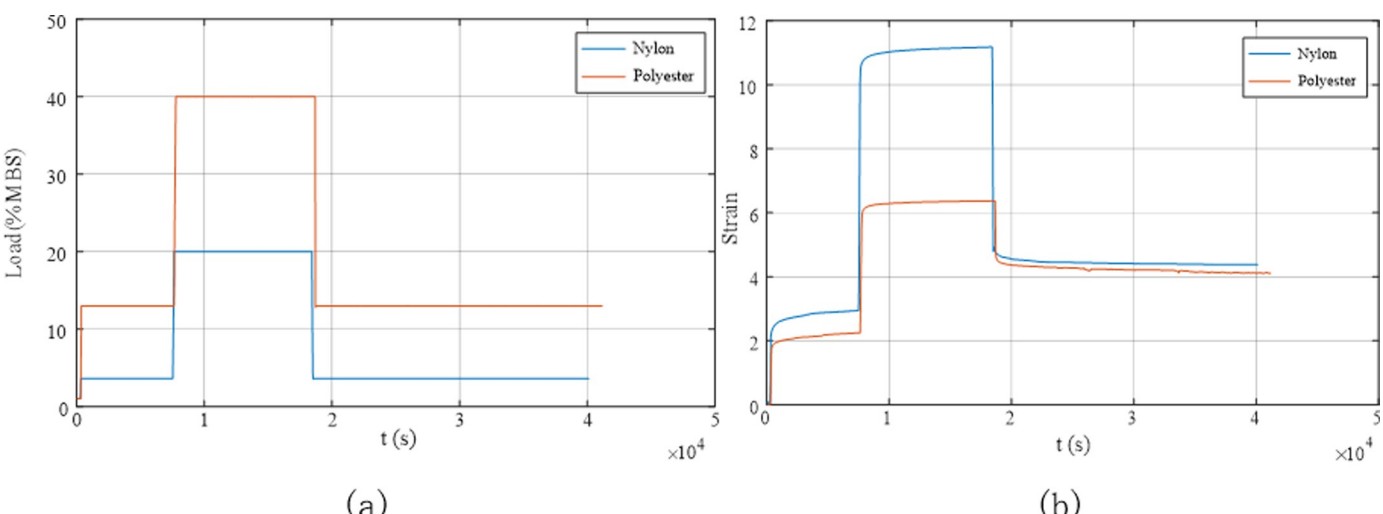

**Fig 16. Deformation of the rope under different load conditions during the preloading test.** (a) Load variation curve; (b) Deformation curve.

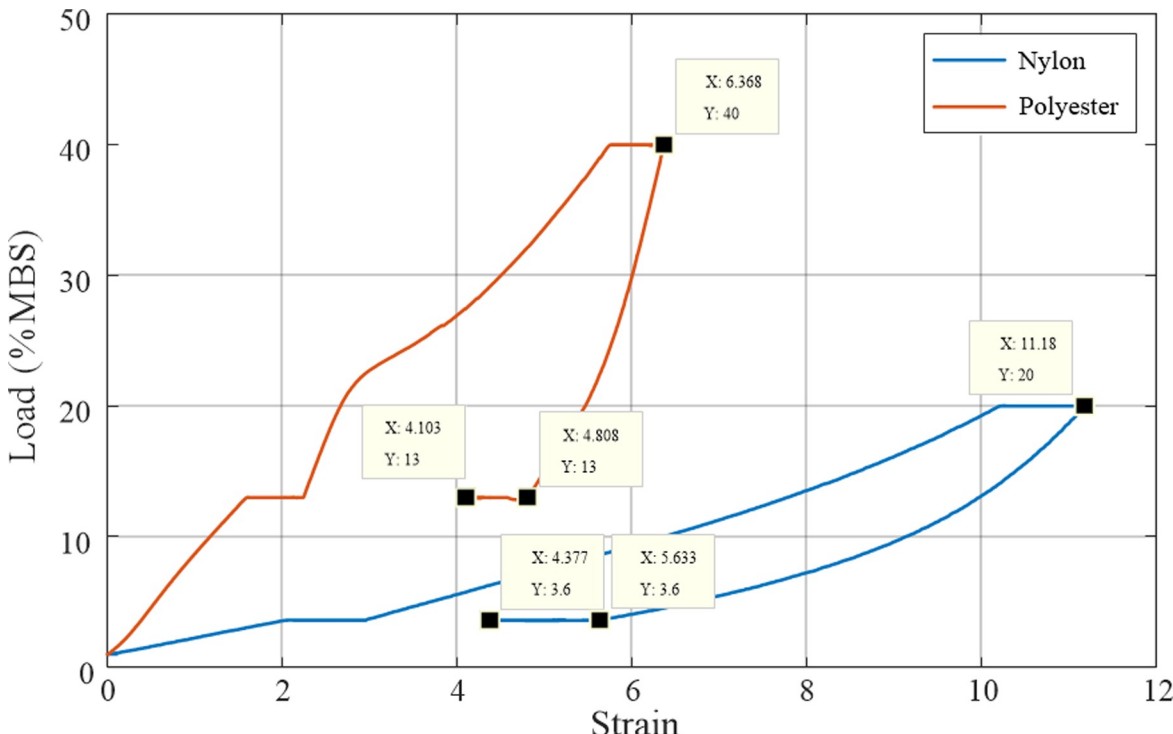

**Fig 17. Stress–strain curves for the synthetic fibre ropes during the preloading test.**

of the polyester rope was 5.207%, and that of the nylon rope was 6.338%. Following 200 min of static holding, the strain of the polyester and nylon ropes recovered to 4.729% and 5.250%, respectively, indicating reversible elongation of 0.478% and 1.088%, respectively.

As different tensile loads correspond to specific creep coefficients, the elongation of the rope was recorded at 1-, 10-, and 100-min intervals (assuming zero relative strain at 1 min), with the relevant values presented in Table 3.

The relative strain gradually increased, and the rope deformation also increased as more time elapsed. The experimental data presented in Table 3 were processed using MATLAB software, and the linear regression results are shown in Fig 20. As the applied load increased, the slopes of the regression curves for the polyester rope were 0.180, 0.170, and 0.205, and for the nylon rope, they were 0.215, 0.230, and 0.280, representing known rope creep rates of 30%, 45%, and 60% MBS, respectively.

Fig 20 indicates that the magnitude of the rope creep coefficient was positively correlated with the tension load—the greater the tension, the greater the rope's creep coefficient, with the maximum creep coefficient observed at a load of 60% MBS.

Based on the stiffness test data and by substituting tension, strain, and the fitted creep coefficient $C$ into Eq (8), the static stiffness equation for the initially installed ropes under loads of

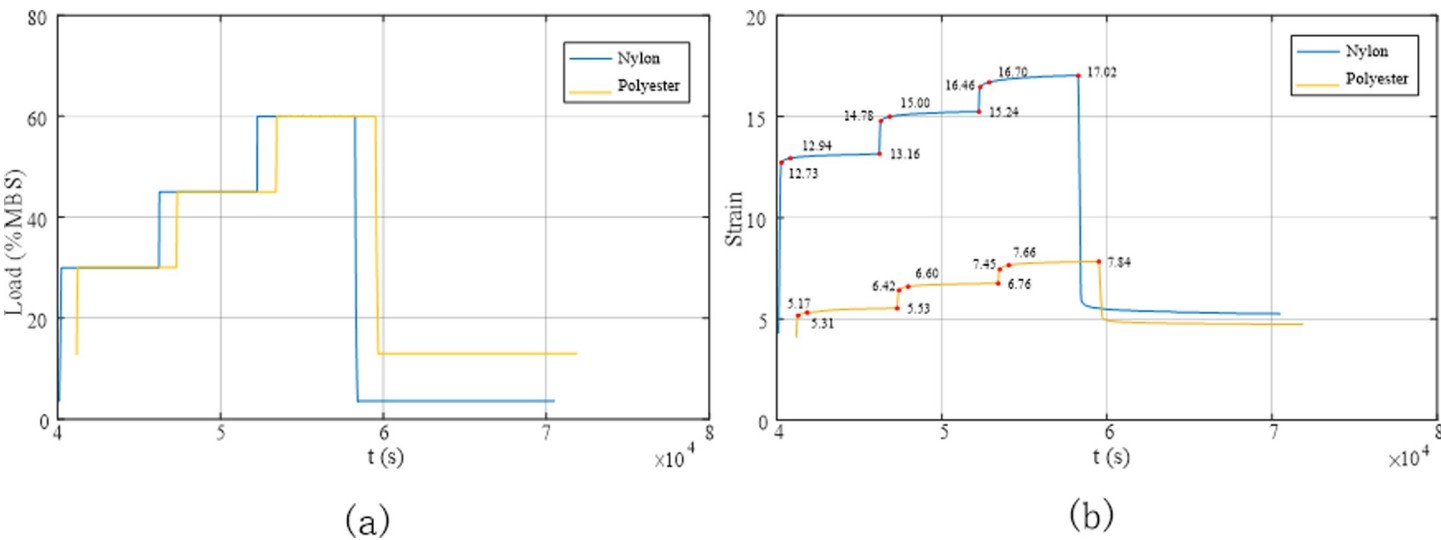

**Fig 18. Deformation of the polyester and nylon ropes under different loads during the initial installation test.** (a) Load variation curve; (b) Deformation curve.

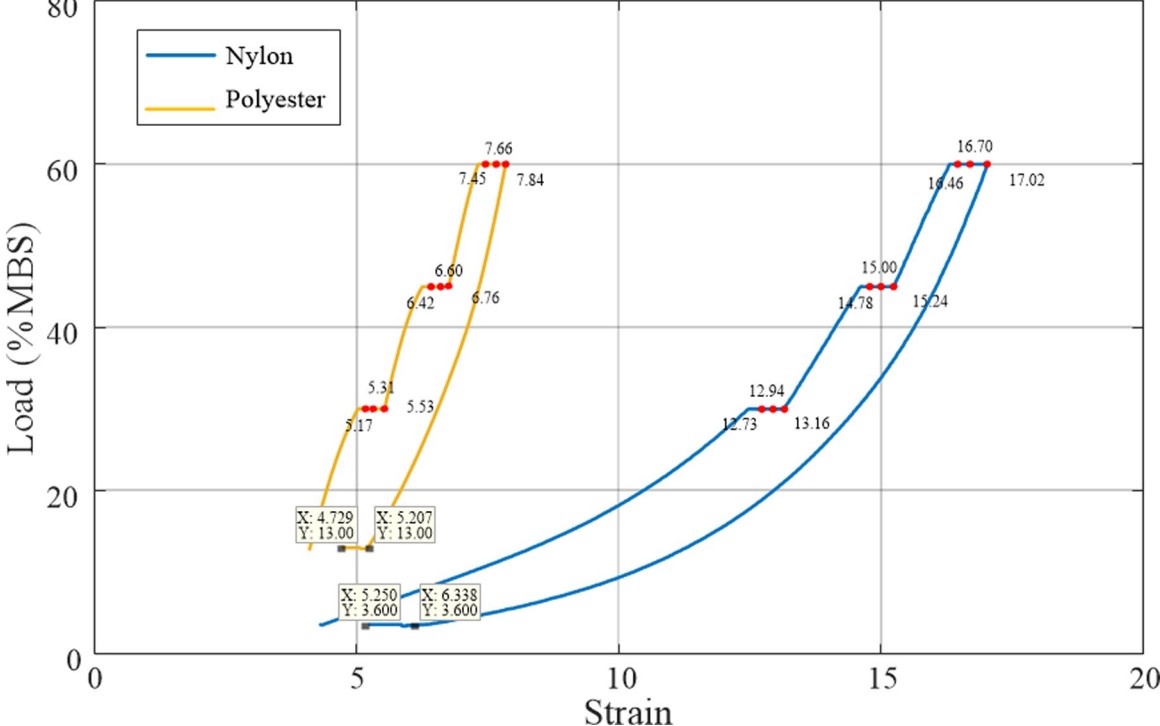

**Fig 19. Stress–strain curve of the synthetic fibre rope in the initial installation test.**

**Table 3. Creep data from the static stiffness testing of the initial rope installation.**

| Load (% MBS) | Time (t) | Lg(t) | Polyester rope strain (%) | Nylon rope strain (%) | Relative strain of polyester rope (%) | Relative strain of nylon rope (%) |
|---|---|---|---|---|---|---|
| 30 | 1 | 0 | 5.17 | 12.73 | 0 | 0 |
|  | 10 | 1 | 5.31 | 12.94 | 0.14 | 0.21 |
|  | 100 | 2 | 5.53 | 13.16 | 0.36 | 0.43 |
| 45 | 1 | 0 | 6.42 | 14.78 | 0 | 0 |
|  | 10 | 1 | 6.60 | 15.00 | 0.18 | 0.22 |
|  | 100 | 2 | 6.76 | 15.24 | 0.34 | 0.46 |
| 60 | 1 | 0 | 7.45 | 16.46 | 0 | 0 |
|  | 10 | 1 | 7.66 | 16.70 | 0.21 | 0.24 |
|  | 100 | 2 | 7.84 | 17.02 | 0.41 | 0.56 |

30%, 45%, and 60% MBS can be solved using

$$
\begin{aligned}
30\%MBS \rightarrow &\quad
\begin{aligned}
&\text{polyester } K_{rs} = 17/[1.427 + 0.180\ lg(t)]\\
&\text{nylon}\quad K_{rs} = 26.4/[8.783 + 0.215\ lg(t)]
\end{aligned}\\
45\%MBS \rightarrow &\quad
\begin{aligned}
&\text{polyester } K_{rs} = 32/[2.657 + 0.170\ lg(t)]\\
&\text{nylon}\quad K_{rs} = 41.4/[10.863 + 0.230\ lg(t)]
\end{aligned}\\
60\%MBS \rightarrow &\quad
\begin{aligned}
&\text{polyester } K_{rs} = 47/[3.737 + 0.205\ lg(t)]\\
&\text{nylon}\quad K_{rs} = 56.4/[12.643 + 0.280\ lg(t)]
\end{aligned}
\end{aligned}
\tag{9}
$$

Table 4 lists the calculated results for the static stiffness of the initially installed ropes. The findings show that the creep rate of the polyester ropes was smaller than that of the nylon ropes, with a larger static stiffness. This indicates that polyester ropes exhibit less deformation and possess a more stable structure. Therefore, polyester ropes are relatively safe and more reliable for deep-sea mooring applications.

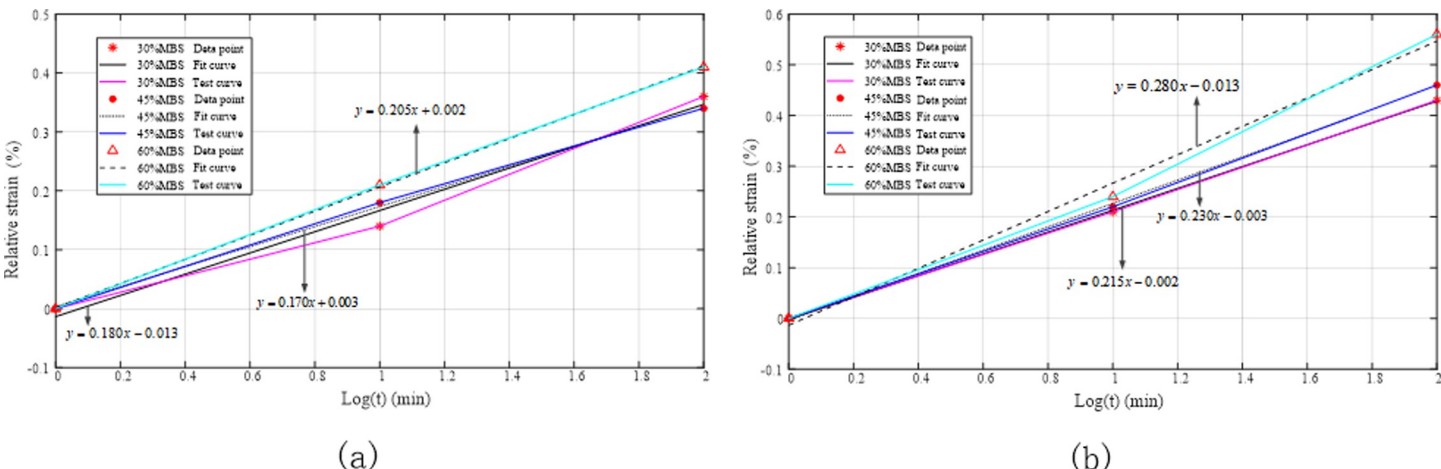

**Fig 20. Fitting of creep coefficients for quasi-static rope stiffness during the initial installation phase.** (a) Polyester rope; (b) Nylon rope.

**Table 4. Calculation results for the static stiffness of the initially installed ropes.**

| Load (% MBS) | Material | Creep rate | $K_{rs}$ (1 min) | $K_{rs}$ (10 min) | $K_{rs}$ (100 min) |
|---|---|---|---|---|---|
| 30 | polyester | 0.180 | 11.913 | 10.579 | 9.513 |
| | nylon | 0.215 | 3.006 | 2.934 | 2.866 |
| 45 | polyester | 0.170 | 12.044 | 11.319 | 10.677 |
| | nylon | 0.230 | 3.811 | 3.732 | 3.656 |
| 60 | polyester | 0.205 | 12.577 | 11.923 | 11.333 |
| | nylon | 0.280 | 4.461 | 4.364 | 4.272 |

## 5.2. Quasi-static ageing rope stiffness test

The test was conducted on the test samples that passed the quasi-static rope stiffness test during the initial installation phase described in Section 4.1. Using periodic dynamic loads to simulate rope ageing, the pretension was increased from the initial tension (13% MBS for polyester and 3.6% MBS for nylon) to 65% MBS and maintained for 100 min. Subsequently, 1000 cycles of dynamic loading were applied continuously, with tension ranging from 35% to 65% MBS and cycles ranging from 12 to 35 s. Finally, after the last cycle, the tension was reduced to the initial tension, and the load applied to the rope was maintained for another 100 min to complete the ageing process. The axial force and strain variation over time are displayed in Fig 21, which clearly shows the static and dynamic loading modes used in the ageing rope test. The purpose of the periodic dynamic loading was to simulate the rope after a period of use, whereas the static loading was used to study the stiffness and elongation characteristics of the ageing rope.

Fig 22 illustrates the stress–strain curve of the ageing rope in the quasi-static stiffness test. The deformation of the rope under static loading was isolated to analyse the creep and determine the creep rate by recording the elongation of the rope, as shown in Fig 22(B). The strain of the polyester rope after unloading was 5.419%, and that of the nylon rope was 5.800%. Following 200 min of static rest, the strain of the polyester and nylon ropes recovered to 5.002% and 4.819%, respectively. Therefore, the reversible elongations of the polyester and nylon

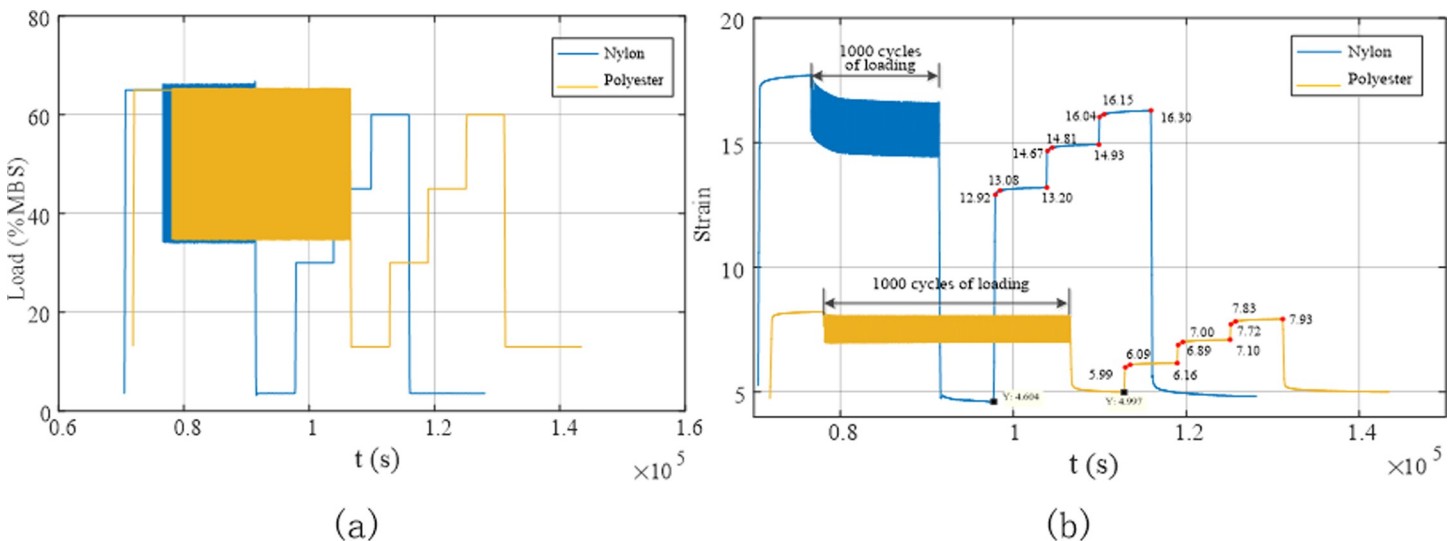

**Fig 21. Deformation of the ageing rope under different load conditions in the quasi-static stiffness test.** (a) Load change curve; (b) Strain curve.

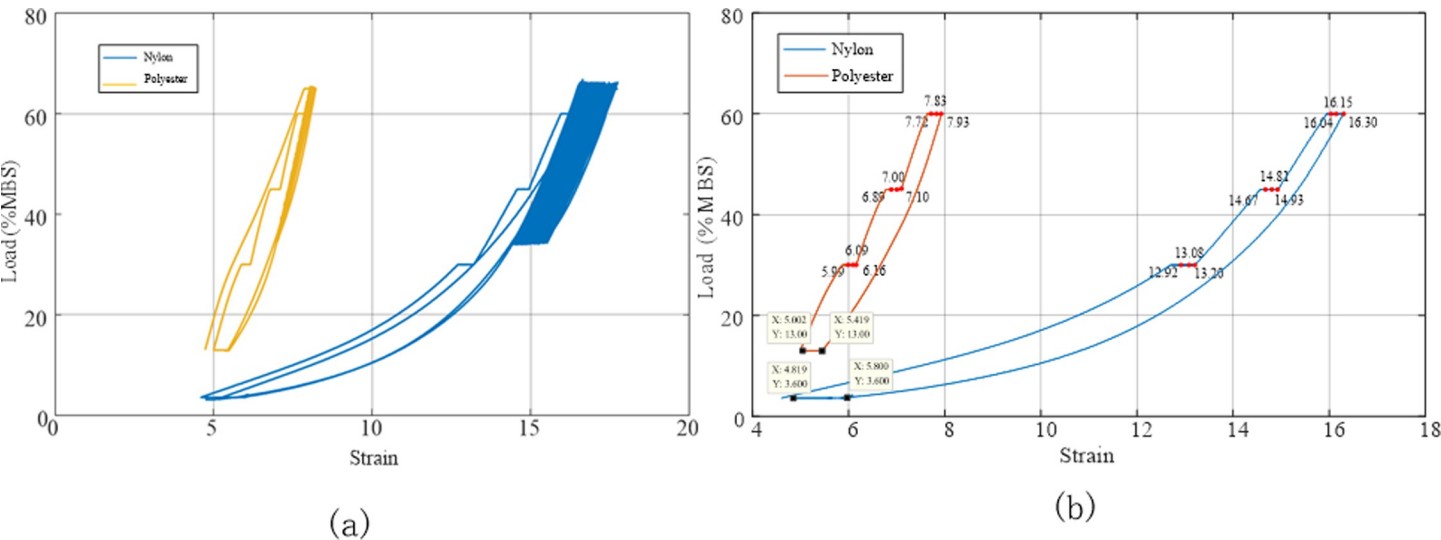

**Fig 22. Stress–strain curves of the ageing synthetic fibre ropes in the ageing test.** (a) Static and dynamic load; (b) Static load phase.

ropes in this stage were 0.417% and 0.981%, respectively (slightly lower than that of the initial installation stage).

The elongation of the ageing rope was recorded at 1, 10, and 100 min (assuming zero relative strain at 1 min). The relevant values are presented in Table 5. Although the strain gradually increased with time, the relative strain sizes tended to be consistent because the ageing ropes had undergone sufficient running-in, and their structures were stable. Fig 23 also confirms this conclusion. The best-fit equations obtained via linear regression analysis for the polyester ropes under 30%, 45%, and 60% MBS loads were coincident, with slopes of 0.085, 0.105, and 0.105, respectively, and for the nylon ropes, the slopes were 0.140, 0.130, and 0.130, respectively, all approaching a constant value.

By substituting tension, strain, and creep rate $C$ into Eq (8), the static stiffness formula of the aged rope can be obtained as

$$
\begin{aligned}
30\%MBS \rightarrow \quad & \text{polyester } K_{rs} = 17/[1.163 + 0.085 \, lg(t)] \\
& \text{nylon} \quad\;\; K_{rs} = 26.4/[8.596 + 0.140 \, lg(t)] \\[4pt]
45\%MBS \rightarrow \quad & \text{polyester } K_{rs} = 32/[2.103 + 0.105 \, lg(t)] \\
& \text{nylon} \quad\;\; K_{rs} = 41.4/[10.326 + 0.130 \, lg(t)] \\[4pt]
60\%MBS \rightarrow \quad & \text{polyester } K_{rs} = 47/[2.933 + 0.105 \, lg(t)] \\
& \text{nylon} \quad\;\; K_{rs} = 56.4/[11.696 + 0.130 \, lg(t)]
\end{aligned}
\tag{10}
$$

Table 6 presents the calculated results for the static stiffness of the aged rope. A comparison with Table 4 indicates that the static stiffness of the aged rope was generally greater than that of the initially installed rope. Moreover, as the load time increased, the change in the dynamic stiffness of the rope decreased. This demonstrated that the aging rope was more fully worn in, effectively eliminating the inherent deformation of the structure and improving the mechanical characteristics of the rope.

Fig 24 illustrates the linear plot of the derived static stiffness equation for the synthetic fibre ropes. The load in the test was equivalent to the force exerted on the ropes in an actual marine

**Table 5. Creep data for the static stiffness test of aged ropes.**

| Load (% MBS) | Time (t) | Lg (t) | Polyester rope strain (%) | Nylon rope strain (%) | Relative strain of polyester rope (%) | Relative strain of nylon rope (%) |
|---|---|---|---|---|---|---|
| 30 | 1 | 0 | 5.99 | 12.92 | 0 | 0 |
|  | 10 | 1 | 6.09 | 13.08 | 0.10 | 0.16 |
|  | 100 | 2 | 6.16 | 13.20 | 0.17 | 0.28 |
| 45 | 1 | 0 | 6.89 | 14.67 | 0 | 0 |
|  | 10 | 1 | 7.00 | 14.81 | 0.11 | 0.14 |
|  | 100 | 2 | 7.10 | 14.93 | 0.21 | 0.26 |
| 60 | 1 | 0 | 7.72 | 16.04 | 0 | 0 |
|  | 10 | 1 | 7.83 | 16.15 | 0.11 | 0.11 |
|  | 100 | 2 | 7.93 | 16.30 | 0.21 | 0.26 |

environment. The horizontal axis represents the duration of action, which facilitates the prediction of static stiffness during the rope production design phase. The figure shows that the static stiffness increased as the applied load increased and decreased as the duration of action increased. Under the same load, the stiffness of the polyester rope was greater than that of the nylon rope, and the static stiffness of the aged rope was greater than that of the initially installed rope.

The permanent mooring systems of FOWTs are generally designed with a lifespan of 25 years, and their main failure modes are divided into tensile cyclic failure and creep failure. However, owing to the significant creep characteristics of HMPE, this may shorten its expected lifespan to reach the 25-year design life. Additionally, its poor economics means that it does not hold an advantage in FOWT mooring systems. In static stiffness tests conducted on polyester and nylon, whether for newly installed ropes or ropes that have undergone ageing treatment, under the application of the same force (including static and cyclic loads), the strain and creep rate of nylon ropes are higher than those of polyester ropes. This indicates that nylon is softer than polyester, more prone to deformation, and has a relatively lower stiffness (consistent with the test results illustrated in Fig 24). Therefore, it can be concluded that polyester ropes are more suitable for application in FOWT mooring systems than nylon ropes.

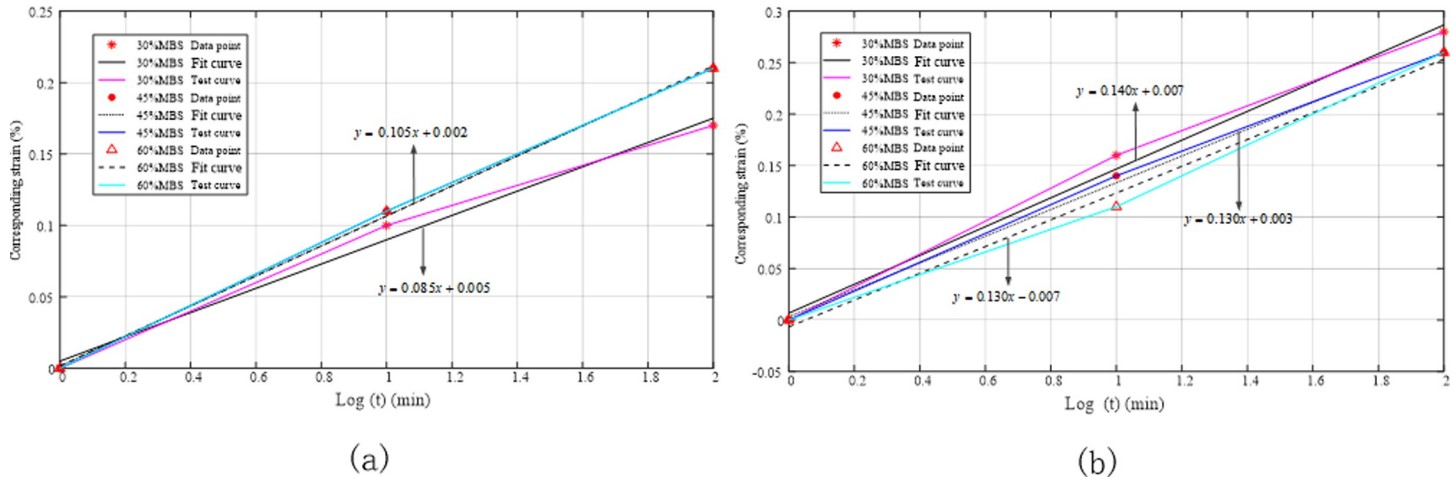

**Fig 23. Best-fit lines for the static stiffness creep coefficient during the ageing stage.** (a) Polyester rope; (b) Nylon rope.

**Table 6. Calculated results for the static stiffness of the aged rope.**

| Load (% MBS) | Material | Creep rate | $K_{rs}$ (1 min) | $K_{rs}$ (10 min) | $K_{rs}$ (100 min) |
|---|---|---|---|---|---|
| 30 | polyester | 0.085 | 14.617 | 13.622 | 12.753 |
|  | nylon | 0.140 | 3.071 | 3.022 | 2.974 |
| 45 | polyester | 0.105 | 15.216 | 14.493 | 13.835 |
|  | nylon | 0.130 | 4.009 | 3.959 | 3.911 |
| 60 | polyester | 0.105 | 16.025 | 15.471 | 14.954 |
|  | nylon | 0.130 | 4.822 | 4.769 | 4.717 |

## 6. Conclusions

This article mainly analysed the mechanical properties of synthetic fibre ropes composed of different materials through experiments, focusing on the wear resistance of fibre ropes and the stiffness characteristics under quasi-static conditions, and the conclusions are given below.

The abrasion test results on yarns demonstrated significant differences in the impact of twist levels on the abrasion resistance of various fibre materials. Specifically, for PET and PA fibres, excessively high twist levels led to mutual damage between fibres, reducing the abrasion resistance of the yarns. By contrast, HMPE fibres exhibited lower sensitivity to changes in twist owing to their superior tensile strength. Simultaneously, dry and wet conditions had a significant impact on the abrasion resistance of yarns. PET and HMPE fibre ropes demonstrated higher abrasion resistance under wet conditions. However, the presence of hydrophilic polar groups in the molecular structure of PA caused a significant decrease in abrasion resistance when wet. Furthermore, studies discovered that under dry conditions, marine lubricants did not significantly enhance yarn abrasion resistance. However, in a wet environment, they could considerably prolong the frictional breakage cycle of the yarn, thereby improving its abrasion resistance. We conclude that synthetic fibre ropes made of PET and HMPE materials are a better choice for FOWT mooring systems.

During the quasi-static stiffness tests on fibre ropes composed of PET and PA materials, it was observed that under small loads, the reversible elongation of PET and PA ropes during the preconditioning phase was approximately 0.705% and 1.256%, respectively. As the load

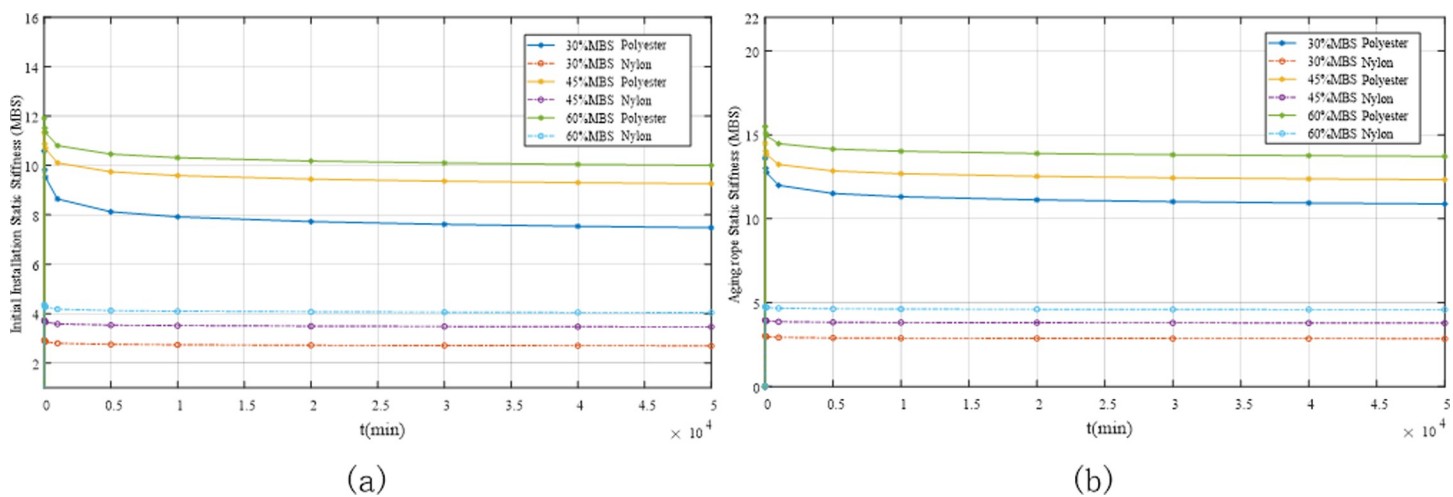

**Fig 24. Design chart of the static stiffness of the fibre ropes.** (a) Initially installed rope static stiffness; (b) Aged rope static stiffness.

continued, the reversible elongation of the ropes during the initial installation and ageing phases correspondingly decreased. This indicated that although the axial force caused damage to the internal structure of the ropes, after sufficient running-in and stabilization, both the elongation and reversible elongation tended to a constant value. An empirical formula for quasi-static stiffness based on the creep coefficient was established using a fitting function of relative strain versus time to calculate the creep coefficient. The study found that the magnitude of the creep coefficient increased with the applied force, and the creep coefficient of PA was greater than that of PET. Additionally, the static stiffness of the ropes showed a linear relationship with the load application time, with the static stiffness of the ropes after ageing being higher than that of the ropes during the initial installation, and the stiffness of PA ropes.

A thorough analysis reveals that PET and HMPE materials demonstrate exceptional wear resistance in fibre ropes. Upon closer examination of the tensile properties and stiffness characteristics of these ropes, it becomes evident that PET ropes exhibit less reversible elongation and possess greater stiffness than PA ropes. Considering these mechanical performance metrics, PET and HMPE synthetic fibre ropes clearly possess relative advantages, offering robust support for the mooring systems of FOWTs. Nevertheless, the creep characteristics of HMPE material are more pronounced, and its cost-effectiveness is less than optimal, which restricts its use in FOWT mooring systems. The results indicate that PET synthetic fiber ropes, with their exceptional wear resistance, high stiffness, and negligible creep, are well-suited for floating wind turbine mooring systems, providing improved safety and reliability.

While this study has yielded valuable insights, it is not without limitations, particularly in the scope of synthetic fiber material selection, which may have restricted the identification of superior alternatives. To further advance the mechanical properties and weather resistance of synthetic fiber ropes, future research should explore the development and application of novel high-performance synthetic materials. These materials are expected to exhibit enhanced strength, wear resistance, and weather resistance, addressing the demands of larger-scale, deeper-water floating offshore wind turbine mooring systems. Additionally, future investigations should focus on the nonlinear behavior of mooring materials, as well as relaxation and tension phenomena during mooring operations, with the goal of optimizing mooring system design and improving overall safety and reliability.

## Supporting information

**S1 Table. Fiber rope mechanical test data.** Include all the raw data used in the experiments described in this document.
(RAR)

## Author Contributions

**Conceptualization:** Hailei Dong, Chiate Chou.

**Data curation:** Hailei Dong, Chiate Chou, Hangyu Li.

**Investigation:** Ji Zeng.

**Methodology:** Ji Zeng, He Zhang, Bowen Jin, Hailei Dong, Chiate Chou.

**Project administration:** Chiate Chou, Hangyu Li.

**Resources:** Hailei Dong, Hangyu Li.

**Software:** Bowen Jin.

**Supervision:** Ji Zeng, He Zhang.

**Validation:** Ji Zeng.

**Visualization:** He Zhang.

**Writing – original draft:** Ji Zeng, He Zhang.

**Writing – review & editing:** Ji Zeng, He Zhang, Bowen Jin.

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
