## [Decision Letter · Decision Letter 0]

15 Sep 2024

PONE-D-24-36084Material selection for floating offshore wind turbine ropes based on wear resistance and stiffnessPLOS ONE

Dear Dr. Zhang,

Thank you for submitting your manuscript to PLOS ONE. After careful consideration, we feel that it has merit but does not fully meet PLOS ONE’s publication criteria as it currently stands. Therefore, we invite you to submit a revised version of the manuscript that addresses the points raised during the review process.

We look forward to receiving your revised manuscript.

Kind regards,

Zeashan Hameed Khan, Ph.D.

Academic Editor

PLOS ONE

Journal Requirements:

3. We are unable to open your Supporting Information file Figures.rar. Please kindly revise as necessary and re-upload.

Reviewers' comments:

Reviewer's Responses to Questions

**Comments to the Author**

1. Is the manuscript technically sound, and do the data support the conclusions?

Reviewer #1: Partly

Reviewer #2: Partly

2. Has the statistical analysis been performed appropriately and rigorously? 

Reviewer #1: N/A

Reviewer #2: Yes

3. Have the authors made all data underlying the findings in their manuscript fully available?

Reviewer #1: No

Reviewer #2: Yes

4. Is the manuscript presented in an intelligible fashion and written in standard English?

Reviewer #1: Yes

Reviewer #2: No

5. Review Comments to the Author

Reviewer #1: 1. It is hard to review since the figures are in the supplemental files not within the manuscript.

2. Remove extra information from the abstract.

3. Literature is not rich at all.

4. I hardly see proper citations and references for the used equations.

5. The results needs further discussion, do not only describe what you see.

6. The conclusion section should include contributions in past tense.

7. Future recommendations should be addressed as well as limitations.

Reviewer #2: Review of the Research Paper

The research presented in this manuscript titled: “Material selection for floating offshore wind turbine ropes based on wear resistance and stiffness.” has significant potential for publication. The authors have taken on an important topic, which could contribute valuable insights to the field. However, there are areas where improvements are necessary to fully unlock the paper's potential and make the findings more accessible and robust.

1. Abstract: The abstract lacks the inclusion of key data and values from the experimental or analytical results. It is common practice to briefly summarize significant findings or numerical results in the abstract to provide a quick overview of the study's outcomes. The authors should include specific data points, performance metrics, or any significant findings from the tests conducted.

2. Title and Material Selection: The title mentions "material selection," but the study does not appear to provide a comprehensive comparison or analysis of the materials in question (e.g., nylon and polyester ropes). While some discussion on the materials' performance exists, it is crucial to focus on the selection criteria and justification for choosing these materials as benchmarks.

3. Literature Review: The number of reviewed papers is quite low. A more thorough examination of the existing literature is encouraged to establish the context for the research. Additionally, in many cases, references are grouped together, which can detract from a focused discussion of each source. It is better to reference individual papers, highlighting their relevance and contribution to the research.

4. Coherence Between Title and Work: The paper lacks coherence between the title and the presented research. The focus on quasi-static stiffness tests is clear, but the connection to material selection, particularly in relation to Floating Wind Offshore Turbines (FWOTs), is not well established. The title should either be adjusted to reflect the primary focus of the work, or the research should be expanded to include a more comprehensive comparison of material selection for FWOTs.

5. Quasi-static Stiffness Tests: The authors present quasi-static stiffness tests on nylon and polyester ropes but fail to provide a comparative analysis or draw conclusions regarding material selection for FWOTs. It is strongly recommended that the authors conclude by analyzing previous data and justifying the selection of the superior material for FWOT applications.

6. Limitations in Measuring Yarn Angles (Lines 154-155): The authors briefly mention challenges in measuring angles between yarns, but the actual difficulties faced by the team are not discussed in detail. It is important to elaborate on the specific limitations encountered, whether they are technical, methodological, or equipment-related, to provide context for the challenges faced during the study.

7. Marine Oil Agents (Lines 216, 232): In multiple sections, the paper refers to "marine oil agents" without specifying which ones are being discussed. The authors should provide the necessary specifications, such as the chemical composition or brand of the agents, to improve clarity and reproducibility.

8. Equation 4: Equation 4 is left underdeveloped. The relations between the variables are not fully explained. It is important to clearly define the functional relationships between these variables and their influence on the quasi-static stiffness. Providing detailed explanations will enhance the readers' understanding of the mechanics behind the model.

9. Equation 5: Equation 5 is incomplete as it lacks an equal sign. This needs to be corrected to properly represent it as an equation.

10. Best Fit Equations (Line 429): The authors mention the formation of "best fit equations" but do not elaborate on the methodology or criteria used for determining these equations. A brief explanation of how the equations were derived and why they are considered the best fit would provide greater transparency and credibility to the analysis.

11. Figure Explanation (Fig 6(a), Fig 9, Fig 24): In Figure 6(a), the number 2418 is used in relation to twist, but this is not explained. The same number appears again in Figures 9 and 24. The authors should clarify the significance of 2418 and its connection to the data presented. Additionally, explain how this value relates to the observed twist behavior in the ropes.

12. Number of Figures: The number of figures in the paper is quite large. The authors should consider reducing the number of figures by combining related ones using multi-plot or another method. This would make the visual data more concise and improve the overall presentation without losing important insights.

By addressing these points, the paper will become more coherent, clear, and in line with best practices for scientific writing and presentation.

6. PLOS authors have the option to publish the peer review history of their article (what does this mean?). If published, this will include your full peer review and any attached files.

Reviewer #1: No

Reviewer #2: No

---

## [Author Response · Author response to Decision Letter 0]

4 Dec 2024

Dear Editors and Reviewers:

Thank you for your letter and for the reviewers’ comments concerning our manuscript entitled “Study on the mechanical behaviour of synthetic fibre ropes in floating offshore wind turbine mooring systems”（PONE-D-24-36084）. Those comments are all valuable and very helpful for revising and improving our paper. We have studied reviewer’s comments carefully and have made revision which marked in red in the paper. The main corrections in the paper and the responds to the reviewer’s comments are as flowing:

Reviewer #1:

1. The review’s comment: It is hard to review since the figures are in the supplemental files not within the manuscript.

The authors’ answer:

Thank you for your valuable feedback. We are well aware of your concerns regarding placing data in the appendix rather than in the main text. This issue has been properly addressed, and we have integrated the collection of charts into the main text to present the information more clearly and facilitate the review process.

2. The review’s comment: Remove extra information from the abstract.

The authors’ answer:

Thank you for your suggestion. We have revised the abstract to make it more concise, clear, and to highlight the core content.

3. The review’s comment: Literature is not rich at all.

The authors’ answer:

Thank you for your suggestion. We have made modifications to the paper content and have found relevant institutions to assist with polishing. The relevant proof is as follows:

4. The review’s comment: I hardly see proper citations and references for the used equations.

The authors’ answer:

Thank you for your suggestion. We have recognized the deficiencies in the citation and references of the paper and have made corrections and additions to the formulas in the article.

5. The review’s comment: The results needs further discussion, do not only describe what you see.

The authors’ answer:

Thank you for your suggestion. We have re-described and supplemented the experimental results.Please refer to lines 240 to 249, 259 to 269, 269 to 289, 509 to 519 of the paper, and the conclusion section.

6. The review’s comment: The conclusion section should include contributions in past tense.

The authors’ answer:

Thank you for your suggestion. We have adjusted the tense of the conclusion section accordingly based on your proposal.

7. The review’s comment: Future recommendations should be addressed as well as limitations.

The authors’ answer:

Thank you for your suggestion. We have rephrased and modified the conclusion section.

Reviewer #2:

1. The review’s comment: Abstract: The abstract lacks the inclusion of key data and values from the experimental or analytical results. It is common practice to briefly summarize significant findings or numerical results in the abstract to provide a quick overview of the study's outcomes. The authors should include specific data points, performance metrics, or any significant findings from the tests conducted.

The authors’ answer: 

Thank you for your feedback. We have rephrased the abstract of the paper, and the revised abstract includes a concise summary of the experimental content and key findings to provide a quick overview of the research results.

2. The review’s comment: Title and Material Selection: The title mentions "material selection," but the study does not appear to provide a comprehensive comparison or analysis of the materials in question (e.g., nylon and polyester ropes). While some discussion on the materials' performance exists, it is crucial to focus on the selection criteria and justification for choosing these materials as benchmarks.

The authors’ answer: 

Thank you for your valuable feedback. We have recognized the inadequacies in the paper's title. This paper primarily focuses on experimental research on wear and stiffness of related mooring ropes, aiming to meet the requirements of mooring. Therefore, after considering your suggestion, we have adjusted the title of the paper accordingly to ensure its consistency with the research content.

3. The review’s comment: Literature Review: The number of reviewed papers is quite low. A more thorough examination of the existing literature is encouraged to establish the context for the research. Additionally, in many cases, references are grouped together, which can detract from a focused discussion of each source. It is better to reference individual papers, highlighting their relevance and contribution to the research.

The authors’ answer:

Thank you for your valuable feedback. We have rewritten the literature review section of the paper, reorganized the references, and increased the amount of literature read. We have cited more individual papers and emphasized the contributions of scholars' research.

4. The review’s comment: Coherence Between Title and Work: The paper lacks coherence between the title and the presented research. The focus on quasi-static stiffness tests is clear, but the connection to material selection, particularly in relation to Floating Wind Offshore Turbines (FWOTs), is not well established. The title should either be adjusted to reflect the primary focus of the work, or the research should be expanded to include a more comprehensive comparison of material selection for FWOTs.

The authors’ answer:

We sincerely thank you for your valuable suggestions. We have recognized the deficiencies in the paper's title and have made appropriate modifications. At the same time, we have re-articulated the conclusion section to ensure that the presentation of the research findings is clearer and more understandable. Please refer to the "Title" and "Conclusion" sections of the paper for details.

5.The review’s comment: Quasi-static Stiffness Tests: The authors present quasi-static stiffness tests on nylon and polyester ropes but fail to provide a comparative analysis or draw conclusions regarding material selection for FWOTs. It is strongly recommended that the authors conclude by analyzing previous data and justifying the selection of the superior material for FWOT applications.

The authors’ answer:

We sincerely thank you for your valuable suggestions. Your questions have played a significant role in improving our paper, and we have added the relevant conclusive content into the main text, which can be found on lines 509 to 519.

6. The review’s comment: Limitations in Measuring Yarn Angles (Lines 154-155): The authors briefly mention challenges in measuring angles between yarns, but the actual difficulties faced by the team are not discussed in detail. It is important to elaborate on the specific limitations encountered, whether they are technical, methodological, or equipment-related, to provide context for the challenges faced during the study.

The authors’ answer:

Thank you for your valuable feedback. We have thoroughly elaborated on the limitations encountered when measuring yarn angles, and the specific details can be found between lines 195 to 204 in the main text of the paper.

7. The review’s comment: Marine Oil Agents (Lines 216, 232): In multiple sections, the paper refers to "marine oil agents" without specifying which ones are being discussed. The authors should provide the necessary specifications, such as the chemical composition or brand of the agents, to improve clarity and reproducibility.

The authors’ answer:

Thank you for your valuable feedback. We have added the materials on the use of marine oil agents to the paper to ensure clarity and reproducibility of the experiments. Specific references can be found on lines 269 to 275.

8. The review’s comment: Equation 4: Equation 4 is left underdeveloped. The relations between the variables are not fully explained. It is important to clearly define the functional relationships between these variables and their influence on the quasi-static stiffness. Providing detailed explanations will enhance the readers' understanding of the mechanics behind the model.

The authors’ answer:

Thank you for your valuable feedback. We have provided a more detailed explanation of the derivation process, which can be found between lines 324 to 332.

9. The review’s comment: Equation 5: Equation 5 is incomplete as it lacks an equal sign. This needs to be corrected to properly represent it as an equation.

The authors’ answer:

We have revised equation 5 in the original paper; details can be found in equation 7.

10. The review’s comment: Best Fit Equations (Line 429): The authors mention the formation of "best fit equations" but do not elaborate on the methodology or criteria used for determining these equations. A brief explanation of how the equations were derived and why they are considered the best fit would provide greater transparency and credibility to the analysis.

The authors’ answer:

We sincerely thank you for your valuable feedback. During the data fitting process, we used MATLAB software to import data and utilized its automatic fitting feature. This allowed us to leverage MATLAB's powerful capabilities and advanced algorithms to analyze the data and determine the fitting curve or function that best reflects the characteristics of the data. This method helps us to more accurately grasp the relationships and trends within the data, thereby providing strong support for our decision-making and research in related fields.

11. The review’s comment: Figure Explanation (Fig 6(a), Fig 9, Fig 24): In Figure 6(a), the number 2418 is used in relation to twist, but this is not explained. The same number appears again in Figures 9 and 24. The authors should clarify the significance of 2418 and its connection to the data presented. Additionally, explain how this value relates to the observed twist behavior in the ropes.

The authors’ answer:

We sincerely appreciate the valuable feedback you have provided. Indeed, we have identified an issue in the PET fiber test condition 18: at the fixed knotting point, the yarn experienced breakage, resulting in a measured cycle count of only 2418. In response, we have updated the relevant details in the main text, which you can refer to on lines 217 to 223.

12. The review’s comment: Number of Figures: The number of figures in the paper is quite large. The authors should consider reducing the number of figures by combining related ones using multi-plot or another method. This would make the visual data more concise and improve the overall presentation without losing important insights.

The authors’ answer:

Thank you for your valuable feedback. We have made every effort to improve the icon issue by merging the original figures 9 and 10. To ensure the complete presentation of the experimental results, no adjustments have been made to the other charts.

---

## [Decision Letter · Decision Letter 1]

13 Dec 2024

PONE-D-24-36084R1Study on the mechanical behaviour of synthetic fibre ropes in floating offshore wind turbine mooring systemsPLOS ONE

Dear Dr. Zhang,

Thank you for submitting your manuscript to PLOS ONE. After careful consideration, we feel that it has merit but does not fully meet PLOS ONE’s publication criteria as it currently stands. Therefore, we invite you to submit a revised version of the manuscript that addresses the points raised during the review process.

We look forward to receiving your revised manuscript.

Kind regards,

Zeashan Hameed Khan, Ph.D.

Academic Editor

PLOS ONE

Journal Requirements:

Additional Editor Comments:

The paper has been improved significantly, however, some suggestions are as follows:

Define all the abbreviations on their first use.

It is recommended to use high resolution images.

What are the future recommendations for this research?

Reviewers' comments:

Reviewer's Responses to Questions

**Comments to the Author**

1. If the authors have adequately addressed your comments raised in a previous round of review and you feel that this manuscript is now acceptable for publication, you may indicate that here to bypass the “Comments to the Author” section, enter your conflict of interest statement in the “Confidential to Editor” section, and submit your "Accept" recommendation.

Reviewer #1: (No Response)

Reviewer #2: All comments have been addressed

2. Is the manuscript technically sound, and do the data support the conclusions?

Reviewer #1: (No Response)

Reviewer #2: Yes

3. Has the statistical analysis been performed appropriately and rigorously? 

Reviewer #1: (No Response)

Reviewer #2: Yes

4. Have the authors made all data underlying the findings in their manuscript fully available?

Reviewer #1: (No Response)

Reviewer #2: Yes

5. Is the manuscript presented in an intelligible fashion and written in standard English?

Reviewer #1: (No Response)

Reviewer #2: Yes

6. Review Comments to the Author

Reviewer #1: (No Response)

Reviewer #2: Dear Editor,

I am pleased to see that the authors have significantly improved the manuscript by addressing all of my comments comprehensively, point by point. I would like to express my appreciation for their efforts and the enhancements made to the overall quality of the paper. I am now satisfied with the current form of the article and would like to submit my acceptance.

However, I have a few minor recommendations that, if considered, could further strengthen the publication:

1. The title has been improved, but it could be refined further for linguistic clarity and to make it more engaging and reader-friendly.

2. While the results and discussions are much clearer now, a brief mention of the study's broader implications or real-world applications in the conclusion could enhance the manuscript's impact.

7. PLOS authors have the option to publish the peer review history of their article (what does this mean?). If published, this will include your full peer review and any attached files.

Reviewer #1: **Yes: **Luttfi A. AL-HADDAD

Reviewer #2: No

---

## [Author Response · Author response to Decision Letter 1]

10 Jan 2025

Dear Editors and Reviewers:

Thank you for your letter and for the reviewers’ comments concerning our manuscript entitled “Mechanical behavior of synthetic fiber ropes for mooring floating offshore wind turbines”（PONE-D-24-36084R1）. Those comments are all valuable and very helpful for revising and improving our paper. We have studied reviewer’s comments carefully and have made revision which marked in red in the paper. The main corrections in the paper and the responds to the reviewer’s comments are as flowing:

Additional Editor Comments:

1. The editor’s comment: Define all the abbreviations on their first use.

The authors’ answer: Thank you for your suggestion. We have revised the acronyms; details can be found on line 107 of the article and in Table 1.

2. The editor’s comment: It is recommended to use high resolution images.

The authors’ answer: Thank you for your valuable feedback. The relevant figures have been updated. Please see the main text for reference.

3. The editor’s comment: What are the future recommendations for this research?

The authors’ answer: We sincerely appreciate the valuable feedback you have provided. We have supplemented our outlook for the future in lines 562 to 571 of the article. please refer to the main text for details.

Reviewer #2: 

1. The review’s comment: The title has been improved, but it could be refined further for linguistic clarity and to make it more engaging and reader-friendly.

The authors’ answer: Thank you for your valuable suggestion. We have recognized the deficiencies in the paper's title and have made appropriate modifications to the title of the paper.

2. The review’s comment: While the results and discussions are much clearer now, a brief mention of the study's broader implications or real-world applications in the conclusion could enhance the manuscript's impact.

The authors’ answer: Thank you for your valuable feedback. We have added the corresponding content to lines 558-561 of the article. please refer to the main text for details.

---

## [Editor Report · Decision Letter 2]

13 Jan 2025

Mechanical behavior of synthetic fiber ropes for mooring floating offshore wind turbines

PONE-D-24-36084R2

Dear Dr. Zhang,

We’re pleased to inform you that your manuscript has been judged scientifically suitable for publication and will be formally accepted for publication once it meets all outstanding technical requirements.

Kind regards,

Zeashan Hameed Khan, Ph.D.

Academic Editor

PLOS ONE

Additional Editor Comments (optional):

The paper has been further improved after including the limitations of the current research and addition of possible future extensions. Therefore, it can be accepted in the present form.
---

## [Editor Report · Acceptance letter]

27 Jan 2025

PONE-D-24-36084R2 

PLOS ONE

Dear Dr. Zhang, 

I'm pleased to inform you that your manuscript has been deemed suitable for publication in PLOS ONE. Congratulations! Your manuscript is now being handed over to our production team.

Kind regards, 

on behalf of

Dr. Zeashan Hameed Khan 

Academic Editor

PLOS ONE